# In vitro-transcribed antigen receptor mRNA nanocarriers for transient expression in circulating T cells in vivo

N. N. Parayath[1,6], S. B. Stephan[1,6], A. L. Koehne[2], P. S. Nelson[3,4] & M. T. Stephan [1,4,5✉]

Engineering chimeric antigen receptors (CAR) or T cell receptors (TCR) helps create disease-specific T cells for targeted therapy, but the cost and rigor associated with manufacturing engineered T cells ex vivo can be prohibitive, so programing T cells in vivo may be a viable alternative. Here we report an injectable nanocarrier that delivers in vitro-transcribed (IVT) CAR or TCR mRNA for transiently reprograming of circulating T cells to recognize disease-relevant antigens. In mouse models of human leukemia, prostate cancer and hepatitis B-induced hepatocellular carcinoma, repeated infusions of these polymer nanocarriers induce sufficient host T cells expressing tumor-specific CARs or virus-specific TCRs to cause disease regression at levels similar to bolus infusions of ex vivo engineered lymphocytes. Given their ease of manufacturing, distribution and administration, these nanocarriers, and the associated platforms, could become a therapeutic for a wide range of diseases.

[1] Clinical Research Division, Fred Hutchinson Cancer Research Center, Seattle, WA 98109, USA. [2] Translational Pathology, Fred Hutchinson Cancer Research Center, Seattle, WA 98109, USA. [3] Division of Human Biology and Clinical Research, Fred Hutchinson Cancer Research Center, Seattle, WA 98109, USA. [4] Division of Medical Oncology, Department of Medicine, University of Washington, Seattle, WA 98195, USA. [5] Department of Bioengineering and Molecular Engineering & Sciences Institute, University of Washington, Seattle, WA 98195, USA. [6]These authors contributed equally: N. N. Parayath, S. B. Stephan. ✉email: mstephan@fredhutch.org

The efficacy of adoptive T-cell therapies, a powerful modality in which T cells harvested from the patient or a donor are genetically modified to target cancers or infectious agents, is supported by numerous clinical trials showing impressive outcomes[1–4]. However, the complexity and high costs involved in manufacturing a bespoke T-cell product for each patient, rather than preparing a drug in bulk in a standardized form, make it difficult to compete with frontline therapy options such as small molecule drugs or monoclonal antibodies[5]. Most CAR-T and TCR-engineered T cells are currently made by a cumbersome process involving: (i) Leukapheresis to extract T cells from a patient who is connected by two intravenous tubes to an apheresis machine for several hours[6]. This is uncomfortable for the patient, incurs a substantial monetary cost, and ultimately may be rate-limiting for large-scale adoption of autologous T cell; (ii) Activation and transduction of T cells; (iii) Expansion of transduced T cells over an approximately two-week period in a cytokine-supplemented tissue culture medium; (iv) Washing and concentrating the T cells prior to administration. For T-cell products made at central facilities and transported to remote treatment centers, cells have to be cryopreserved; and (v) Quality control release assays are necessary for each batch of CAR-T product. The entire process has to be conducted under environmentally controlled GMP-compliant conditions, which are expensive to maintain and run. Because each CAR-T product is made from starting materials (T cells) from the patient to be treated, there are no economies of scale[7].

In vitro-transcribed (IVT) mRNA has emerged as a disruptive new drug class that can be used to encode therapeutically relevant proteins of interest directly in vivo[8,9]. Synthetic mRNA molecules can be quickly designed and manipulated and mass-produced relatively cost-effectively[10]. Over the past decades, scientists have learned how to optimize mRNA pharmacologically and immunologically to make it more drug-like for clinical applications[11–14].

Here, we explore mRNA as an injectable drug to genetically reprogram circulating T lymphocytes to transiently express disease-specific receptors, thereby bypassing the need to extract and culture lymphocytes from patients (Fig. 1). To protect the therapeutic payload and to precisely target it to T cells, we formulated biodegradable polymeric nanocarriers. We first demonstrate ex vivo that a single nanoparticle application can routinely transfect >70% of cultured T cells with the CD19-specific 1928z CAR (FDA-approved for the treatment of B-cell lymphoma[15]) or with the HBcore18-27 TCR specific for the hepatitis B virus (HBV) core antigen (currently in a Phase I study to treat patients with HBV-related hepatocellular carcinoma; ClinicalTrials.gov Identifier: NCT03634683). Nanoparticle-transfected T cells transiently express these CAR-transgenes or TCR-transgenes on their surface for an average of 7 days. Using orthotopic xenograft mouse models of lymphoma, prostate cancer, and HBV-induced hepatocellular carcinoma, we demonstrate that, when administered periodically, CAR-encoding or TCR-encoding mRNA particles can genetically reprogram circulating T cells to induce antitumor responses with similar efficacies compared to conventional adoptively transferred T cells that have been virally transduced ex vivo.

The insertion of CARs or TCRs into lymphocytes by gene transfer currently occurs outside the patient's body, in specialized manufacturing suites, but both the process of transporting cells to and from the clean room and the gene transfer procedure itself are labor-intense, expensive, and take valuable time. Should engineered T cell therapy reach its promise of extending to diverse populations across a variety of cancer types, the challenges of economics and manufacturing will likely grow.

We demonstrate that a mRNA nanodrug can achieve the power of effective, side-effect-free cell therapy with the convenience of an off-the-shelf drug. Just as for a conventional drug, with this new treatment modality the patient could be easily redosed for as long as medically necessary.

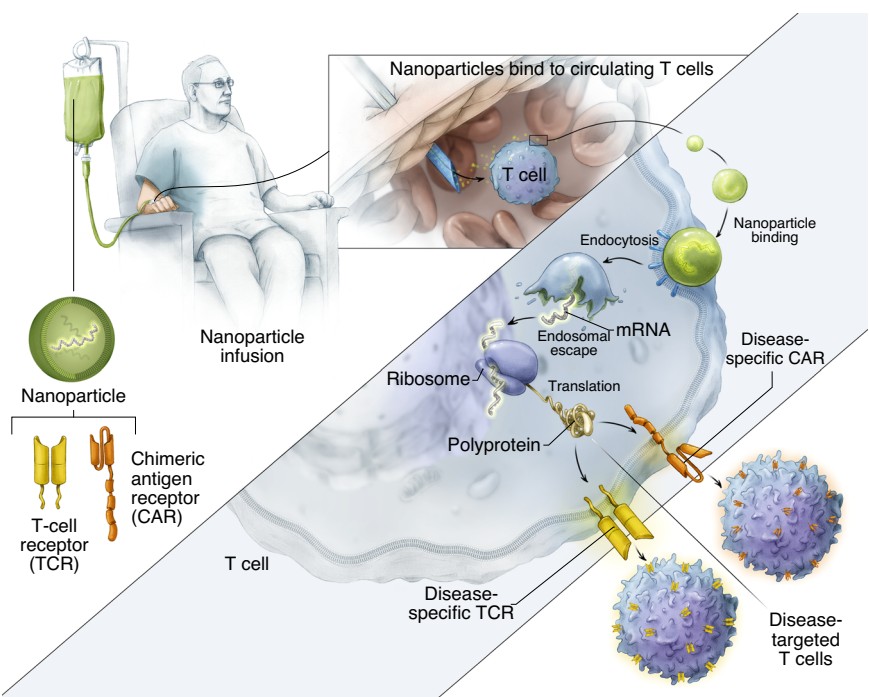

**Fig. 1 Schematic illustrating how we reprogram T cells in situ to express disease-specific CARs or TCRs using IVT mRNA carried by polymeric nanoparticles.** These particles are coated with ligands that target them to cytotoxic T cells, so once they are infused into the patient's circulation they can transfer the transgenes they carry into the lymphocytes and transiently program the cells to express the disease-specific CARs or TCRs on their surfaces.

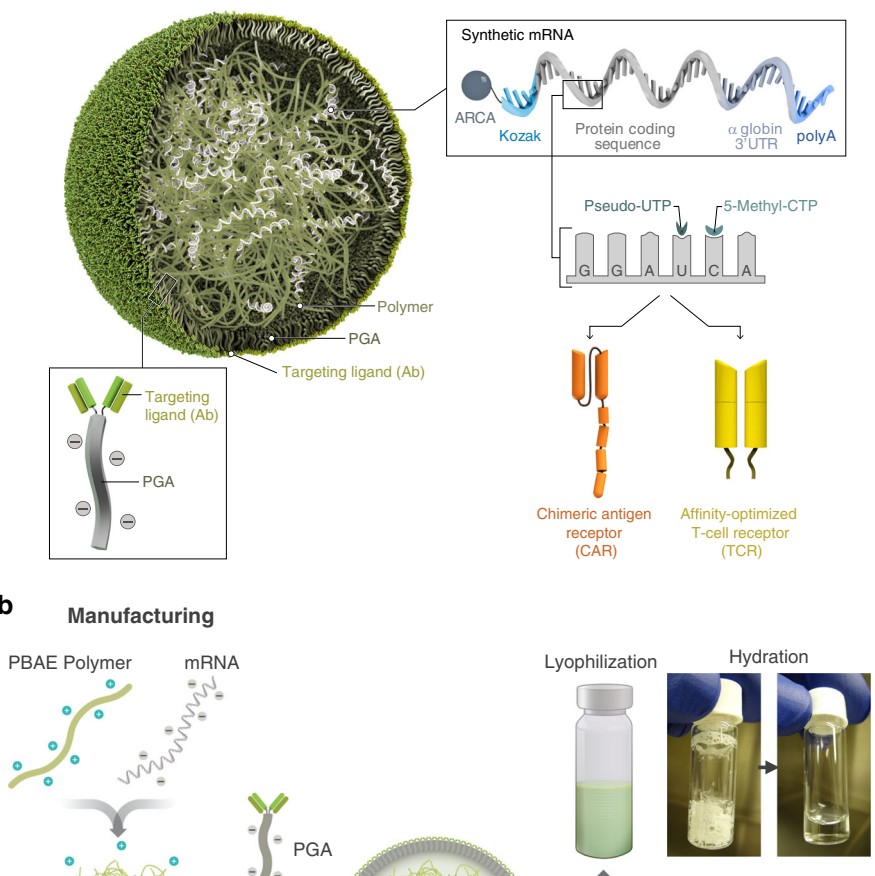

**a** Nanoparticle with synthetic mRNA cargo

**b** Manufacturing

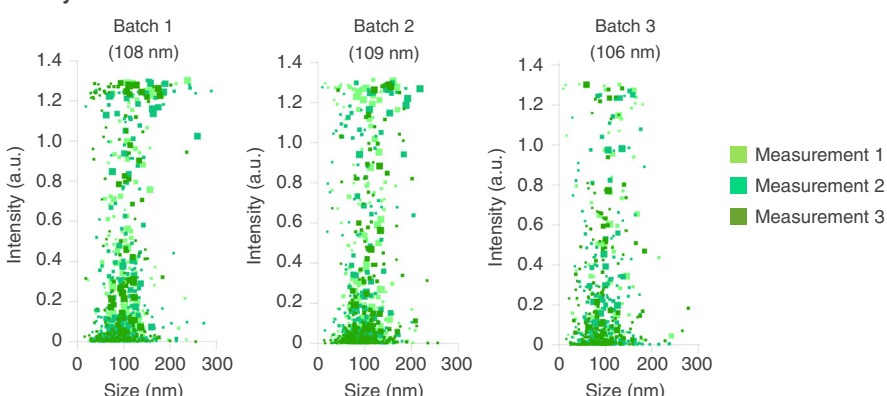

**c** Physicochemcial Characterization

Batch 1 (108 nm)

Batch 2 (109 nm)

Batch 3 (106 nm)

Measurement 1
Measurement 2
Measurement 3

## Results

**MRNA nanocarriers transfect T cells with CAR- or TCR transgenes.** To deliver IVT mRNA encoding disease-specific receptor genes into human lymphocytes, we used a biodegradable poly(ß-amino ester) (PBAE) polymer formulation as a carrier matrix (Fig. 2a). The PBAE-447 polymer we are using in our study to condense mRNA into nanoparticles was originally developed in the Jordan Green laboratory at Johns Hopkins University[16]. Over the past decade, his group and others have extensively characterized the key properties of PBAE[17]. PBAEs enable endosomal escape by undergoing protonation at the lower pH of the endosomal compartment, leading to osmotic pressure buildup due buffering, which causes endosomal disruption. High-throughput combinatorial library screens of PBAEs for nucleic acid delivery have shown that the presence of tertiary amines improves buffering capacity at low pH and facilitate endosomal escape. The ester bonds in the backbone structure of PBAEs undergo hydrolysis in aqueous conditions, making PBAEs less

**Fig. 2 Design and manufacture of lymphocyte-programming nanoparticles. a** Schematic of the T cell-targeted IVT mRNA nanocarrier used in our experiments. To create a reagent that can genetically modify primary T lymphocytes (which are refractory to nonviral transfection methods) simply by contact, we bioengineered polymeric nanoparticles comprised of four functional components: (**i**) surface-anchored targeting ligands that selectively bind the nanoparticles to T cells and initiate rapid receptor-induced endocytosis to internalize them. In our experiments we used anti-CD8 antibodies; (**ii**) a negatively charged coating that shields the nanoparticles to minimize off-target binding by reducing their surface charge. Because it is already widely used in drug delivery platforms, we selected polyglutamic acid (PGA) to accomplish this; (**iii**) a carrier matrix that condenses and protects the nucleic acids from enzymatic degradation while they are in the endosome, but releases them once the particles are transported into the cytoplasm, thereby enabling translation of the encoded protein. For this, we used a biodegradable poly(β-amino ester) (PBAE) polymer formulation that has a half-life between 1 and 7 h in aqueous conditions; and (**iv**) nucleic acids (IVT mRNA) that are encapsulated within the carrier and produce transient expression of the disease-specific CAR or TCR. **b** Diagram describing how we fabricated the nanoparticles. **c** Size distributions, measured using a NanoSight NS300 instrument. The mean diameter ± SD, ζ potential, and mRNA encapsulation ± SD are indicated on the top. $N = 3$ independently manufactured nanoparticle batches.

toxic than other nondegradable cationic polymers, such as PEI, which has been broadly investigated as a nucleic acid delivery vehicle. Cationic PBAE self-assembles into nanocomplexes with anionic nucleic acids via electrostatic interactions (Fig. 2b). The particles were made cell-targeting by coupling an anti-CD8 antibody to polyglutamic acid (PGA), forming a conjugate that was electrostatically adsorbed to the particles. The resulting mRNA nanocarriers can be lyophilized for long-term storage. Prior to use, particles hydrate within seconds following addition of sterile water to restore their original concentration. We used Particle Tracking Analysis (NanoSight NS300, Malvern Panalytical) to characterize the particles manufactured in ten independent batches (Fig. 2c). We found the mean diameter of PbAE/PGA-anti-CD8 nanoparticles to be 106.9 ± 7.2 nm. The ζ potential was 4 ± 2, and mRNA encapsulation, measured with a Qubit RNA HS assay kit, was 90.9 ± 6.2% at the 60:1 PBAE:mRNA ratio we used in our nanoparticle formulation.

We first determined whether adding targeted IVT mRNA nanocarriers to a culture of human lymphocytes can give robust transfection of the cells. To test our technology in clinically relevant system, we loaded nanoparticles with IVT mRNA encoding the leukemia-specific 1928z CAR (Fig. 3a–e). CD19-targeted receptors are the most investigated CAR-T cell product today, with nearly 30 ongoing clinical trials internationally, and three already FDA-approved cancer therapies[18,19]. As a second example, we delivered IVT mRNA encoding a high-affinity HBV-specific TCR (Fig. 3f–j). T-cell therapy of chronic hepatitis B is a novel approach to restore antiviral immunity and cure the infection. The HBcore18–27 TCR specific for the HBV core antigen was isolated from an HLA-A 02.01 donor with resolved HBV infection[20]. For both the 1928z CAR and the HBcore18-27 TCR constructs, we first used real-time quantitative PCR and flow cytometry to measure their expression levels in human T cells following a single nanoparticle transfection. We found that transgene expression peaked 24 h after nanoparticle exposure, followed by a gradual decline (Fig. 3a, f). Notably, only nanoparticles functionalized with T-cell-specific (antiCD8 or antiCD3) antibodies efficiently delivered the transgene, whereas isotype control-functionalized nanoparticles yielded gene expression close to background levels (Supplementary Fig. 1). This translated into high levels of CAR or TCR surface expression, with a maximum on day 2 (75 ± 11% of T cells expressed the 1928z CAR, Fig. 3b, c; and an average of 89 ± 4% of T cells expressed the HBcore18-27TCR, Fig. 3g, h). As expected, receptor expression was transient, and was reduced to 28 ± 6% for the CAR and 26 ± 9% for the TCR after 8 days in culture. We next compared function (killing and cytokine production) of nanoparticle-transfected T cells against T cells engineered with these receptors using viral methods. To demonstrate specificity for tumor antigens we included control groups of T cells transduced either with tumor-irrelevant CAR genes (P28z, targeting the Prostate-Specific Membrane Antigen[21]), or TCR

genes (MSLN-TCR, specific for Mesothelin[22]). Using real-time IncuCyte® live cell assays, we could not measure significant differences in the ability of IVT mRNA-transfected T cells to selectively lyse antigen-positive target cells (Raji lymphoma cells for the 1928z CAR and HepG2 liver cancer cells stably transduced with HBcAg for the HBcore18–27 TCR) (Fig. 3d, i). Also, similar levels of T-cell-secreted effector cytokines were measured in nanoparticle-transfected versus virally transduced T cells (Fig. 3e, j).

**Nanoparticles reprogram host T cells to recognize leukemia.** We next examined whether lymphocyte-targeted IVT mRNA nanoparticles can reprogram circulating T cells in quantities large enough to bring about tumor regression with efficacies that are similar to conventional methods. As a first in vivo test system, we modeled leukemia by inoculating immunodeficient NOD.Cg-Prkdcscid Il2rgtm1Wjl/SzJ (NSG) mice with $1 \times 10^6$ CD19+ Raji cells expressing firefly luciferase. Five days later, mice were reconstituted with $10 \times 10^6$ CD3+ human T cells and received six weekly infusions of nanoparticles loaded with mRNA (50 μg/dose) encoding the 1928z CAR to generate leukemia specificity or control particles loaded with mRNA encoding GFP (Fig. 4a). The weekly nanoparticle administration protocol was chosen based on the kinetics of CAR surface expression we measured ex vivo with IVT mRNA nanoparticles, which showed relevant receptor expression for up to 8 days (Fig. 3b, c). To compare the therapeutic efficacy of nanoparticle infusions with conventional adoptive T cell therapy, we also treated an additional group of mice with a single dose of $5 \times 10^6$ T cells transduced ex vivo with lentiviral vectors encoding the 1928z CAR. This quantity is equivalent to the higher doses of CAR T cells used in current clinical studies, where patients have been treated with up to $1.2 \times 10^7$ CAR T cells per kilogram of body weight[23]. In another administration protocol, we combined adoptive transfer of lentivirally transduced CAR T cells with systemic injections of nanoparticles loaded with control GFP mRNA to determine, if nanoparticle-mediated transfection impairs the antitumor function of T cells. Control mice received either no treatment or were infused with untransduced human effector T cells. Tumor growth was serially quantified by bioluminescence imaging and differences in survival were monitored. We found that adoptive transfer of ex vivo-engineered 1928z CAR-T cells robustly improved survival. Six of ten mice eradicated tumors, and the others showed tumor regression with an average 32-day survival improvement (Fig. 4b, c). This therapeutic benefit achieved with conventional adoptive T-cell therapy was similar to treatments with IVT mRNA nanoparticles programming the same CARs into the lymphocytes in vivo, which achieved tumor eradication in 7 of 10 mice and an average 37-day improvement in survival of the relapsing animals (Fig. 4c). Flow cytometry of peripheral blood 2 days after the first dose revealed that 1928z-carrying

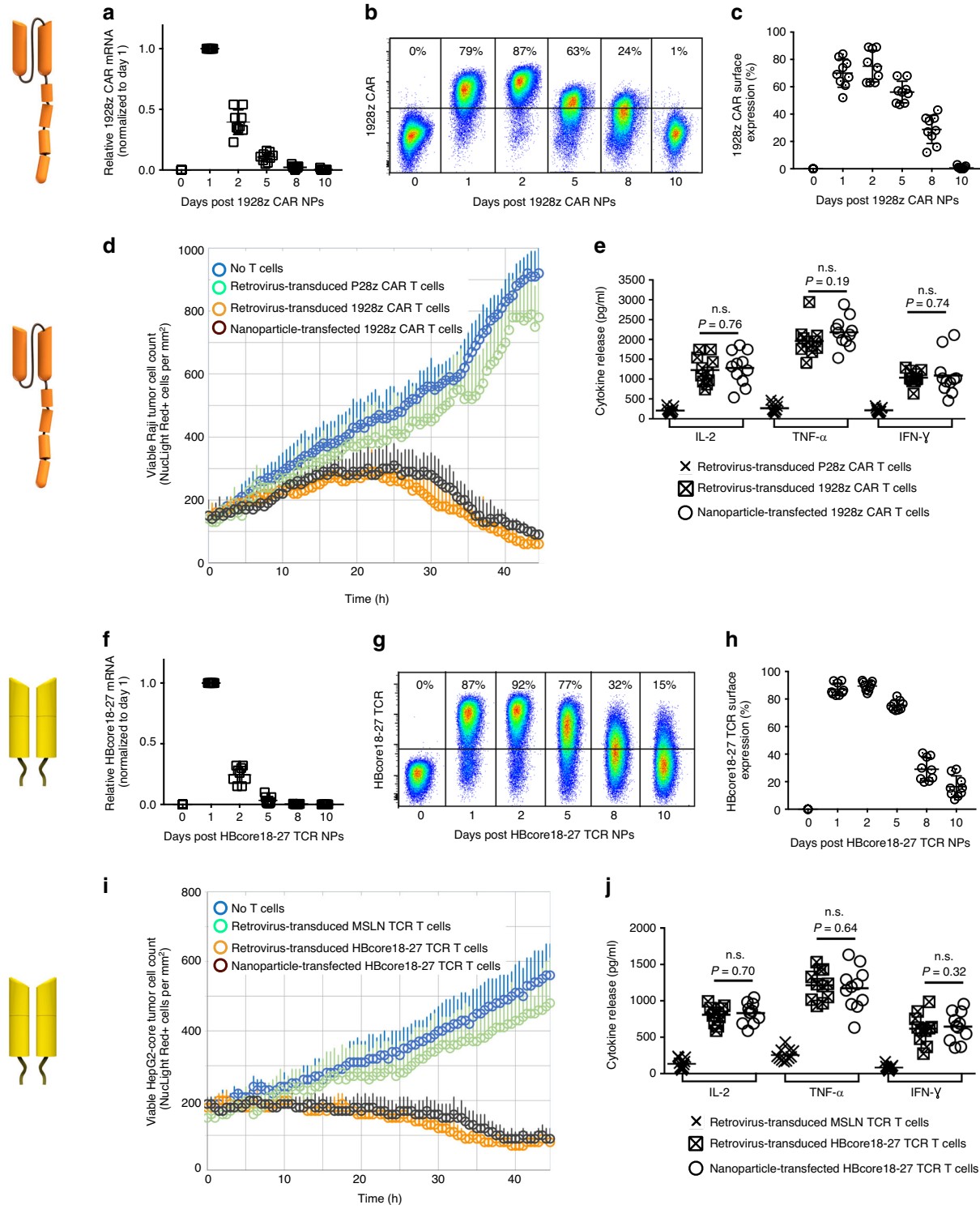

nanoparticles efficiently reprogrammed circulating T cells to recognize leukemia cells (mean 10% CAR+ among CD8+ ± 4.3%, Fig. 4d, e). As expected with mRNA delivery systems, these CARs were transiently expressed for up to 1 week (0.8 ± 0.4% CAR+ CD8+ T cells on day 7). Notably, repeat doses of nanoparticles were as effective as the first injection and achieved an average of 10.7 ± 3.6% gene transfer into host T cells (Fig. 4e). This suggests that, despite its transient nature, IVT mRNA can serve as a platform to achieve persistent in situ CAR expression in host lymphocytes.

**Therapeutic responses in fully immunocompetent hosts**. To examine how exclusively targeting can confine nanoparticle interactions to circulating T cells and how it affects their fates, we employed the fully immunocompetent Ai14 reporter mouse[24]. In this genetically modified model, all cells contain a loxP-flanked STOP cassette preventing transcription of a CAG promoter-driven tdTomato protein. Only cells that are successfully trans-fected with mRNA encoding Cre recombinase (Cre) would excise the loxP-flanked STOP cassette, resulting in permanent tdTomato transcription and subsequent strong, amplified tdTomato

**Fig. 3 IVT mRNA nanocarriers efficiently transfect human T cells with CAR or TCR transgenes.** Isolated human CD8+ T cells were stimulated with beads that are coated with antibodies against TCR/CD3 and co-stimulatory CD28 receptors. Twenty-four hours later, beads were removed and CD8-targeted nanoparticles (NPs) containing either mRNA encoding the leukemia-specific 1928z CAR (**a–e**) or the HBcore18-27 TCR (**f–j**) were mixed into the cell suspension at a concentration of 3 μg of mRNA/$10^6$ cells. **a** qPCR measurements of relative 1928z CAR mRNA expression over time after T cells were exposed to 1928z CAR NPs. Shown are mean values ± SD. $N = 9$ biologically independent samples. **b** Flow cytometry of T cells at the indicated time points after incubation with NPs bearing 1928z CAR-encoding mRNA. **c** Summary plot of in vitro gene transfer efficiencies. Shown are mean values ± SD. $N = 9$ biologically independent samples. **d** In vitro assay comparing cytotoxicity of nanoparticle-transfected vs. retrovirus-transfected T cells against Raji lymphoma cells. T cells were co-cultured with Raji tumor cells at a 5:1 ratio. We used the IncuCyte Live Cell Analysis System to quantify immune cell killing of Raji NucLight Red cells by 1928z-CAR or control (P28z-CAR)-transfected T cells over time. Data are representative of two independent experiments. Each point represents the mean ± s.e.m. pooled from two independent experiments conducted in triplicate. **e** ELISA measurements of IL-2 (at 24 h) and TNF-α and IFN-γ (at 48 h) secretion by transfected cells. Shown are mean values ± SD; two tailed unpaired Student's *t*-test. $N = 9$ biologically independent samples. **f** qPCR measurements of relative HBcore18-27 TCR mRNA expression over time after T cells were exposed to HBcore18-27 TCR NPs. Shown are mean values ± SD. $N = 9$ biologically independent samples. **g, h** Gene transfer efficiencies. **i** Cell killing of HepG2-core NucLight Red cells by HBcore18-27 or control (MSLN-) TCR-transfected T cells over time. T cells were co-cultured with HepG2 tumor cells at a 5:1 ratio. $N = 9$ biologically independent samples. **j** ELISA measurements of cytokine secretion by transfected cells. Shown are mean values ± SD; two tailed unpaired Student's *t*-test.

expression. We first measured whole-organ fluorescence in Ai14 mice following injection of CD3-targeted (or isotype control-functionalized) nanoparticles carrying Cre mRNA. The highest gene expression mediated by nontargeted particles were found in the liver, whilst lymphocyte-targeted nanocarriers induced gene transfer mainly in the spleen, lymph nodes, and thymus (Fig. 5a, b). A detailed flow cytometry analysis of the spleen (Fig. 5c) revealed that CD3-targeted nanoparticles preferentially transfected T cells (8.1 ± 1.9%), without compromising viability (Supplementary Fig. 2). Much lower levels of dtTomato signals were detected in other CD45+ (immune) subtypes, such as macrophages (3.2 ± 1.5%), B cells (1.1 ± 0.9%), neutrophils (0.3 ± 0.2%), and dendritic cells (1.9 ± 0.8%).

Based on these distribution studies, we next tested whether the quantities of mRNA nanoparticle-redirected T cells we measured are sufficient to reduce established cancers in fully immunocompetent hosts. To this end, we infused luciferase-expressing Eμ-ALL01 leukemia cells into albino C57BL/6 mice (to model B cell acute lymphoblastic leukemia in an immunocompetent mouse model[25]) and used bioluminescent imaging to quantify differences in tumor progression between treatment groups (Fig. 6a). Mice either received CD3-targeted nanoparticles delivering mRNA encoding the fully murine 1928z CAR, or GFP control (Supplementary Methods). A third group received no treatment. We found that only infusions of nanocarriers encoding the 1928z CAR effectively controlled leukemia progression (Fig. 6b, c), resulting in an average 26-fold reduced tumor burden after three weeks of therapy compared to GFP controls.

**Therapeutic responses in solid tumors.** To confirm that this technology has relevance not only for the treatment of hematological malignancies, but also for solid tumors, we next investigated the ability of nanoparticles designed to introduce prostate tumor-specific CAR genes into circulating host T cells to induce regression of prostate tumors in mice. Unlike leukemia cells, which express high levels of the CD19 antigen and can easily be accessed by circulating lymphocytes, solid malignancies are heterogeneous and protected[26]. This means that a portion of the tumor cells will evade recognition by the targeting CAR and will be surrounded by immune-suppressing defenses that can render T cells dysfunctional. In fact, co-author Dr. Peter Nelson used whole-genome/transcriptional profiling of 140 prostate cancer metastases to establish that prostate tumor lesions exhibit heterogeneous expression of three key cell-surface proteins (prostate-specific membrane antigen (PSMA), prostate stem cell antigen (PSCA), and receptor tyrosine kinase-like orphan receptor 1 (ROR1) among patients (Fig. 7a). To recapitulate human disease,

we orthotopically transplanted LNCaP C42 prostate carcinoma cells, which exhibit heterogeneous expression of these key cell-surface proteins (Fig. 7b), into the dorsal lobe of the prostate gland of NSG mice (Fig. 7c). To serially monitor tumor burden by bioluminescence imaging, tumor cells were genetically tagged with Firefly luciferase (FLuc). Following orthotopic transplantation, all mice reproducibly developed lesions within three weeks (Fig. 7c, right panel), were reconstituted with human $10 \times 10^6$ CD3+ human T cells, and randomly assigned to the various treatment or control groups (Fig. 7d). We first measured the therapeutic efficacy of systemically injecting tumor-bearing mice with $10^6$ ex vivo-transduced CAR+ T cells specific for the tumor antigen ROR1. We found that even though antiROR1 CAR-T cells did not achieve tumor clearance, the survival rates of the treated mice more than doubled (69 vs. 32 days in the no-treatment control group; Fig. 7d). To determine whether our "off-the-shelf" nanoreagent can achieve similar therapeutic effects, we systemically injected mice weekly with ROR1 CAR transgene-loaded nanoparticles (50 μg mRNA/dose; Fig. 7e). Particle-induced CAR programming extended survival by an average of 40 days compared to untreated controls, which is similar to the survival benefit achieved with conventional adoptive T-cell therapy (Δ mean survival = 3 d, N.s., $P = 0.23$; Fig. 7d, f). Appropriate localization and persistence of T cells is a prerequisite for activity against solid tumors, so we next assessed the frequency of infiltrating ROR1 CAR-T cells into prostate tumors over time. Flow cytometry of LNCaP C42 prostate tumors resected 4 days, 7 days, and 11 days after T-cell transfer revealed that intravenously infused T cells traffic to tumor sites (average 892 ± 295 CAR+ T cells/mg tumor tissue) but do not thrive (only 1.04-fold overall expansion between day 4 and day 11; Fig. 7g, h). Also, in situ-programmed CAR-T cells infiltrate tumors efficiently (average 648 ± 240 CAR+ T cells/mg tumor tissue) and maintain high levels of the CAR transgene before downregulating the receptor (average 91 ± 7 CAR+ T cells/mg tumor tissue on day 7, Fig. 7h). Following an intravenous booster dose of ROR1-CAR-encoding mRNA nanoparticles the same day, the tumor lesions were again infiltrated by freshly reprogrammed peripheral blood T cells (average 1066 ± 225 CAR+ T cells/mg tumor on day 11; Fig. 7h), which recapitulates the oscillating kinetics of mRNA nanoparticle-induced T-cell reprogramming we already observed in our leukemia studies (Fig. 4e).

To determine the cause of failure of both the adoptively transferred T cells and infused mRNA nanocarriers to completely clear the disease, we phenotyped the antigen profile of relapsing prostate tumors by flow cytometry. One of the most common escape strategies seen in cancer is a reduction of target antigen expression because of the selective pressure the CARs create[27,28].

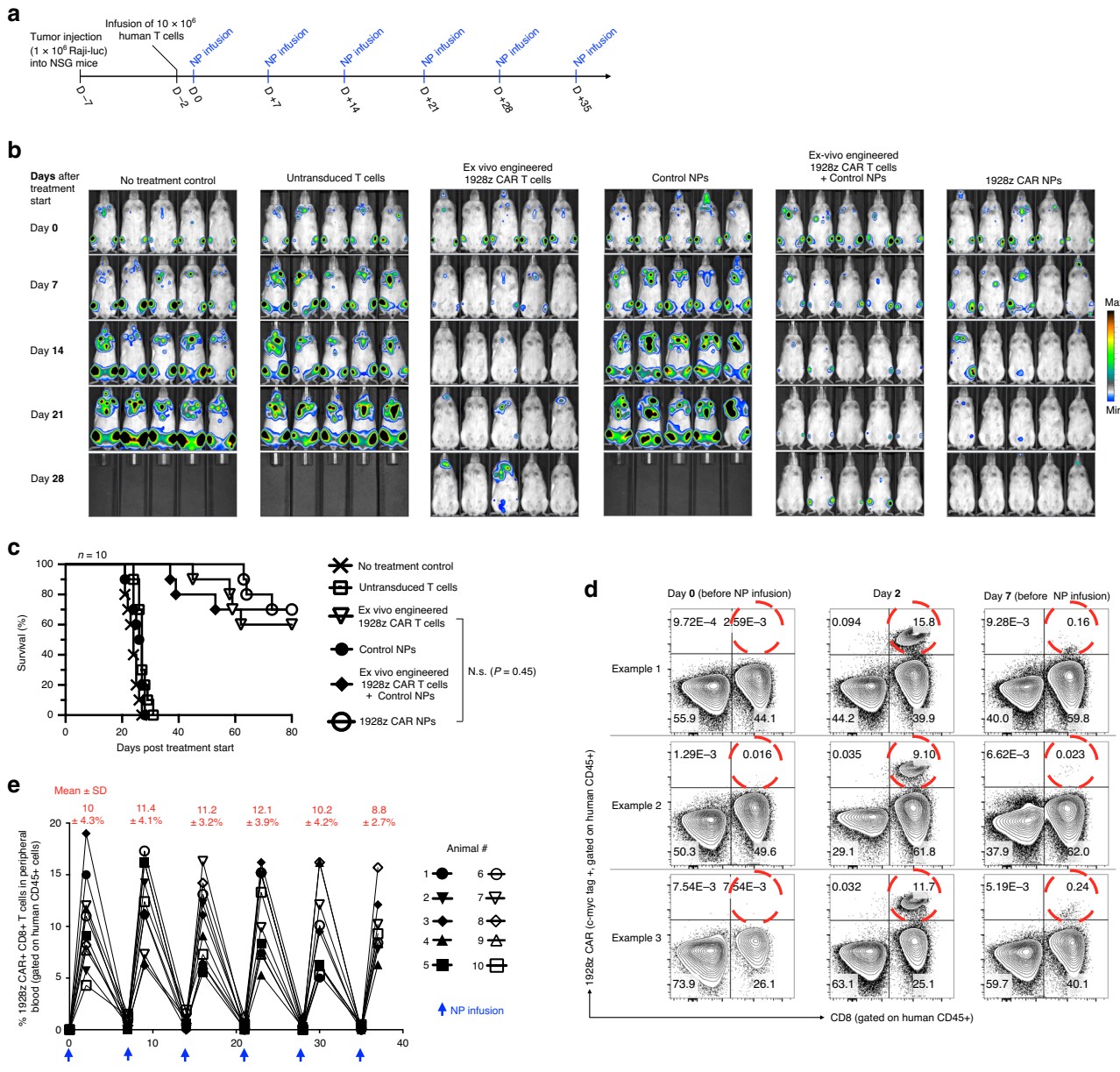

**Fig. 4 Nanoparticle-programmed CAR lymphocytes can cause leukemia regression with efficacies similar to adoptive T-cell therapy. a** Time line and nanoparticle (NP) dosing regimen. **b** Sequential bioimaging of firefly luciferase-expressing Raji lymphoma cells systemically injected into NSG mice. Five representative mice from each cohort ($n = 10$) are shown. **c** Survival of animals following therapy, depicted as Kaplan–Meier curves. Shown are ten mice per treatment group pooled from three independent experiments. ms, median survival. Statistical analysis between the treated experimental and the untreated control group was performed using the Log-rank test; $P < 0.05$ was considered significant. **d** Flow cytometry of peripheral T cells before and after injection of nanoparticles delivering IVT mRNA that encodes the 1928z CAR. The three profiles for each time point shown here are representative of two independent experiments consisting of ten mice per group. **e** Overview graph displaying the percentages of CAR-transfected CD8+ T cells following repeated infusion of 1928z CAR NPs. Every line represents one animal. Shown are ten animals pooled from two independent experiments. Mean transfection efficiencies (±SD) for each time point are shown at the top.

This phenomenon has been reported as a cause of failure in both preclinical and clinical studies, when adoptively transferred T cells specific for only single antigens were used to treat heterogeneous tumors (such as metastatic prostate cancer[29–31]). We found that, in comparison to untreated LNCaP C42 prostate tumors, which express the ROR1 tumor antigen at various levels, CAR-targeted tumors in both treatment groups (adoptively transferred T cells or nanoparticle-programmed T cells) eventually developed ROR1 low/negative immune-escape variants (Fig. 7i).

**In situ programming of HBV-specific T cells**. Gene transfer of IVT mRNA-encoding CARs can only target T cells to antigens located on the surface of cells[32], so the many tumor-antigens or viral-antigens that are intracellular are inaccessible to these receptors[33]. We already demonstrated in vitro that lymphocyte-targeted IVT mRNA nanoparticles can reprogram T cells with engineered TCRs that recognize the intracellular HBV core antigen (HBcAg) in the context of HLA (Fig. 3f–j). Given that over 300 million people worldwide are chronically infected with HBV, with a significant number of these patients developing

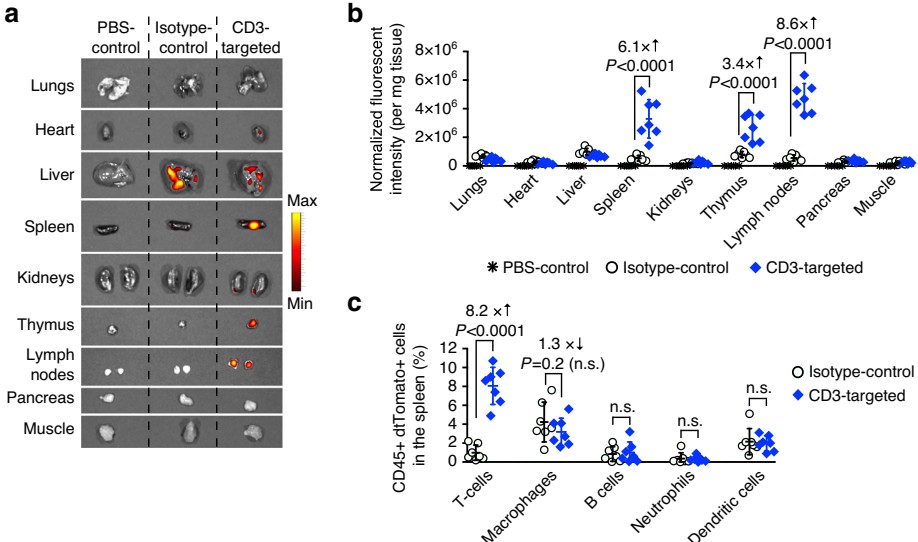

**Fig. 5 Efficient T-cell targeting in immunocompetent mice.** B6.Cg-Gt(ROSA)26Sortm14(CAG-tdTomato)Hze/J (Ai reporter) mice were injected intravenously with three daily doses of nanoparticles loaded with 15 μg mRNA encoding nuclear localization signal (NLS)-Cre. Nanoparticles were targeted to mouse T cells using a full-length anti-CD3 MuIgG2a, or IgG2a isotype control. Both antibodies were designed as LALAPG variants to ablate Fc receptor binding and complement activation. Forty-eight hours after the final injection, organs were collected and whole-organ dtTomato fluorescence was measured using fluorescent IVIS imaging. Single cell suspensions of spleens and blood were labeled with antibodies against various immune cell subtypes and analyzed by flow cytometry. **a** Representative (N = 7, 1 pictured) dtTomato expression in organs under fluorescent IVIS imaging. **b** Quantification of fluorescent signal in each organ. Each symbol indicates one measured organ. Horizontal lines indicate mean values, and error bars represent standard deviation of the mean. Pairwise differences in fluorescent intensities between the groups were statistically analyzed by two tailed unpaired Student's t-test. N = 7 biologically independent samples pooled from two independent experiments. **c** Graph displaying the mean ± SD percent of immune CD45+ dtTomato+ cell types in the spleen. Macrophages (CD45+, CD11b+, MHCII+, CD11c−, Ly6C−/low, Ly6G−), B cells (CD45+, B220+), T cells [CD4+ T cells (CD45+, TCR-β chain+, CD4+, CD8-), CD8+ T cells (CD45+, TCR-β chain+, CD4−, CD8+)], neutrophils (CD45+, CD11b+, MHCII+, CD11c−, Ly6G+), and dendritic cells (CD45+, CD11c+, CD11b−, MHCII+) were measured. Each symbol indicates one mouse. Horizontal lines indicate mean values, and error bars represent standard deviation of the mean. Pairwise differences in fluorescent intensities between the groups were statistically analyzed by two tailed unpaired Student's t-test. N = 7 biologically independent samples.

cirrhosis and liver cancer[34], customizing T-cell products for each patient individually is clearly not feasible. As a first step to implement our IVT mRNA technology for the treatment of this disease, we established a mouse xenograft tumor model of HBV-induced hepatocellular carcinoma (HCC). One million HepG2 cells stably transduced with HBcAg and luciferase were intrahepatically injected after laparotomy. All mice reproducibly developed multifocal lesions within 7 days (Fig. 8a), at which point they were reconstituted with unstimulated human 10 × 10^6 CD3+ human HLA-A*02:01T cells and received two weekly infusions of nanoparticles loaded with mRNA (50 μg/dose) encoding the HBcore18–27 TCR to generate HCC specificity or control particles loaded with mRNA encoding GFP. A third group of mice was treated with a single dose of 10 × 10^6 CD3 T cells (HLA-A*02:01) transduced ex vivo with retroviral vectors encoding the HBcore18–27 TCR, and control mice received no treatment. Four days after the second nanoparticle administration (day 18), livers were isolated to directly quantify tumor burden using bioluminescence imaging, and single-cell suspensions were labeled with HBV C18-27 MHC I Pentamer® to quantify the percentage of HBcore18–27 TCR-expressing T cells. We found that nanoparticle injections programmed sufficient HBcore antigen-specific T cells to induce disease regression and can achieve similar therapeutic effects compared to ex vivo-engineered lymphocytes (13-fold vs. 18.9-fold reduced photon count, respectively, compared to no-treatment control, Fig. 8b, c). Flow cytometry of the dissected livers confirmed equal densities of HBcore18-27 TCR T cells in animals treated with ex vivo-engineered T cells versus in situ programming nanoparticles (Fig. 8d, e).

In conclusion, our results demonstrate that repeated infusions of T-cell-targeted polymer nanocarriers can deliver tumor-specific CARs or virus-specific TCR transgenes into sufficient quantities of host T cells to induce disease regression at levels similar to bolus infusions of ex vivo-engineered lymphocytes.

**T cell-programming nanoparticles are biocompatible.** As a first step toward advancing in situ programming of disease-specific T cells to clinical application, we worked with the Nanoparticle Characterization Laboratory (NCL) at the National Cancer Institute (https://nanolab.cancer.gov/). Systemic administration of nanomedicines has the potential to cause infusion reactions in patients, an adverse response which often delays or halts clinical translation[35]. These reactions can manifest as fevers, chills, rigors, rashes, chest, or back pain, or difficulty in breathing, and, in rare instances, they can be fatal. Identifying the risk of infusion reactions early in the drug development process can help mitigate potential safety concerns once the product reaches clinical trials, saving developers both time and money—and saving patients from potentially dangerous complications.

We used Assay Cascade Protocols, developed by the NCL, which can be indicative of infusion reactions. Specifically, we analyzed the effect of nanoparticles on complement activation (NCL Method ITA-5.2), their hemolytic properties (ITA-1), and their effect on oxidative stress in T-cells (ITA-32). To study these effects in clinically relevant concentrations of nanoparticles, we first calculated the Theoretical Plasma Concentration (TPC)[36], which is the efficacious mouse dose (in our experiments: 50 μg

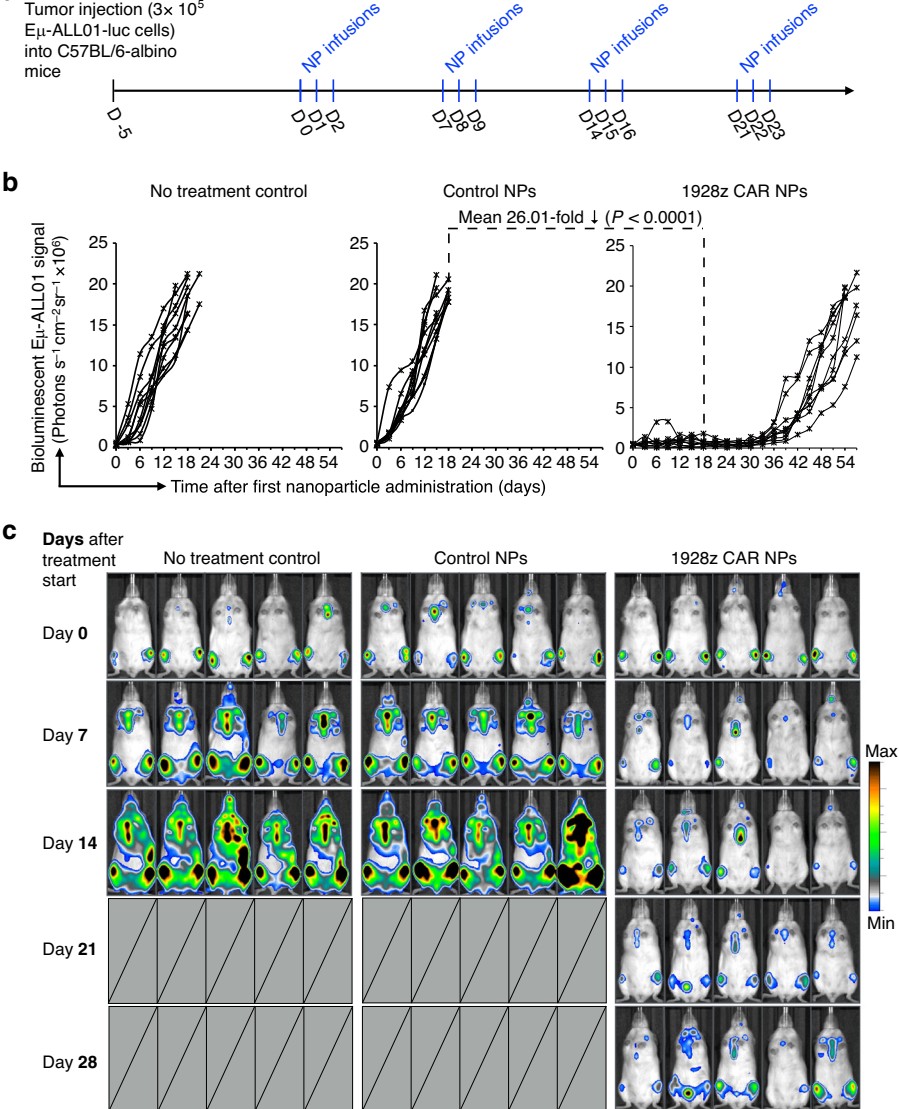

**Fig. 6 Antileukemia response in immunocompetent mice. a** Time line and dosing regimen. **b** Plots of Eμ-ALL01 luciferase signal intensities after nanoparticle injections. Each line represents one animal and each dot reflects its whole animal photon count. Statistical differences were examined by two tailed unpaired Student's *t*-test. Shown are data for ten mice per treatment condition pooled from two independent experiments. **c** Sequential bioimaging of firefly luciferase-expressing Eμ-ALL01 leukemia cells systemically injected into albino C57BL/6 mice. Five representative mice from each cohort (*n* = 10) are shown.

mRNA/dose) scaled to the equivalent human dose (=2.03 μg mRNA/mL blood; see Fig. 9a). To assess the impact of T-cell-targeted mRNA nanoparticles on erythrocytes, we performed hemolysis tests by spectrophotometric measurement of hemoglobin release after exposure to various concentrations of particles. The performance of the hemolysis assay was tested by negative (PBS) and positive (Triton-X) controls. We found that the hemolysis rates for particles at the TPC were lower than 2% (mean 1.21 ± 0.26%, compared to 0.7 ± 0.11% in the PBS control; Fig. 9b), which is defined as non-hemolytic[37]. Nanoparticles also did not induce activation of complement iC3b or Bb, while C4d was slightly above the 2-fold assay threshold at the TPC concentration (mean 2.3 ± 0.13%, compared to 1 ± 0.003% in the PBS control; Fig. 9c). Lastly, we measured mitochondrial oxidative stress as a key determinant of nanoparticle-induced injury, because excessive production of reactive oxygen species (ROS) causes damage to cellular organelles and DNA, eventually leading to cell death[38]. Moreover, another consequence of

excessive ROS generation is activation of cell signaling pathways that stimulate expression of proinflammatory and fibrotic cytokines[39]. We found that T-cell-programming nanoparticles induced only a very modest increase in oxidative stress (mean 3.6 ± 0.2-fold), compared to the PBS control (Fig. 9d).

Guided by the in vitro assessment of T-cell-programming mRNA nanoparticles for their potential to cause infusion reactions, we next conducted a comprehensive toxicity assessment in rodents. The rat is the preferred rodent species to predict human health toxicity outcomes of nucleic acid-based molecular therapies because its metabolic physiology (in particular renal and hepatic function) is closer to that of humans than is the mouse's[40–42]. Sprague Dawley rats (6–8 weeks old) were injected with one dose of nanoparticles carrying 100 μg mRNA, which is the rat equivalent of 50 μg mRNA in mice, based on normalization of dose to body surface area[43]. These experiments were conducted using 1928z CAR nanoparticles. The 1928z CAR recognizes human CD19 but does not cross-react with rat CD19,

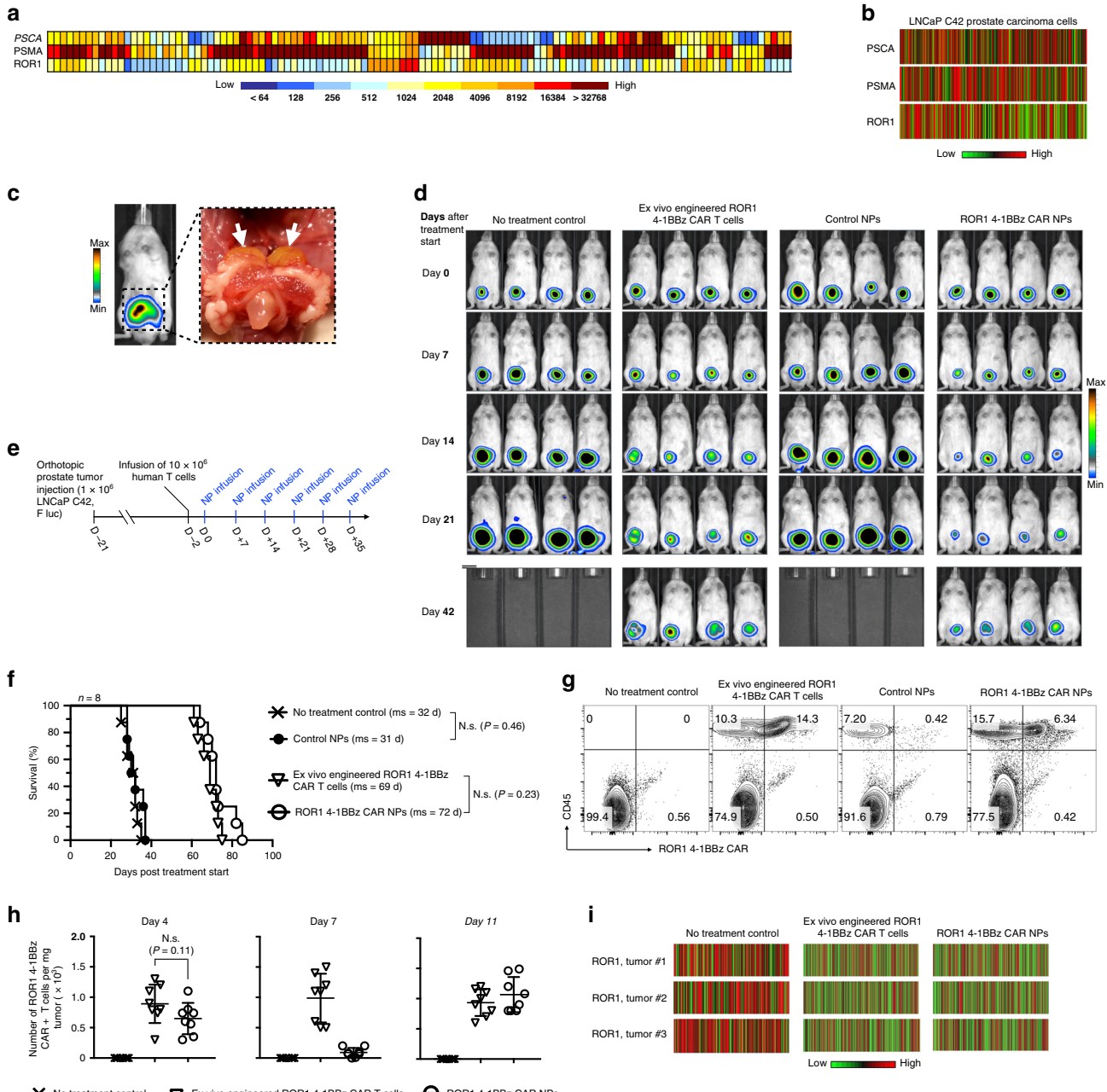

**Fig. 7 IVT-mRNA nanocarriers encoding prostate tumor-specific CARs can improve survival of mice with established disease. a** Heat map of PSCA, PSMA, and ROR1 antigen expression across a panel of 140 prostate cancer metastases showing the diversity of antigen expression. **b** Heat map representation of flow cytometry data showing variability in PSCA, PSMA, and ROR1 expression by LNCaP C42 prostate carcinoma cells. The colors indicate expression levels in 350 randomly chosen cells. **c** Three weeks of postimplantation, LNCaP C42 prostate tumors were visualized by in vivo bioluminescent imaging. A representative photo of established tumors in the dorsal lobes of the prostates (white arrows) is shown on the right. **d** Sequential bioimaging of firefly luciferase-expressing LNCaP C42 prostate carcinoma cells orthotopically transplanted into the prostate of NGS mice. Four representative mice from each cohort ($n = 8$) are shown. **e** Time line and nanoparticle dosing regimen. **f** Survival of animals following therapy, depicted as Kaplan–Meier curves. Shown are eight mice per treatment group pooled from three independent experiments. ms, median survival. Statistical analysis between the treated experimental and the untreated control group was performed using the Log-rank test; $P < 0.05$ was considered significant. N.s. nonsignificant. **g** Multicolor flow cytometry of cells recovered from prostate tumors 11 days after treatment start. Adoptively transferred or in situ-programmed ROR1 CAR+ T cells were identified by positive labeling for CD45 and a c-myc tag incorporated in the receptor. **h** Absolute numbers of ROR1-CAR+ T cells that localized to tumors isolated on day 4, day 7, and day 11 after treatment start. Total cell counts of viable (trypan blue-negative) cells were multiplied by the percentage that was both RO1-CAR and CD45 positive. Shown are mean values ± SD; two tailed unpaired Student's $t$-test. $N = 8$ biologically independent samples pooled from two independent experiments. **i** Flow cytometry quantification of ROR1 antigen expression on LNCaP C42 prostate tumor cells following CAR-T cell therapy or ROR1 4-1BBz CAR NP therapy. Shown are 350 randomly chosen cells pooled from five tumors.

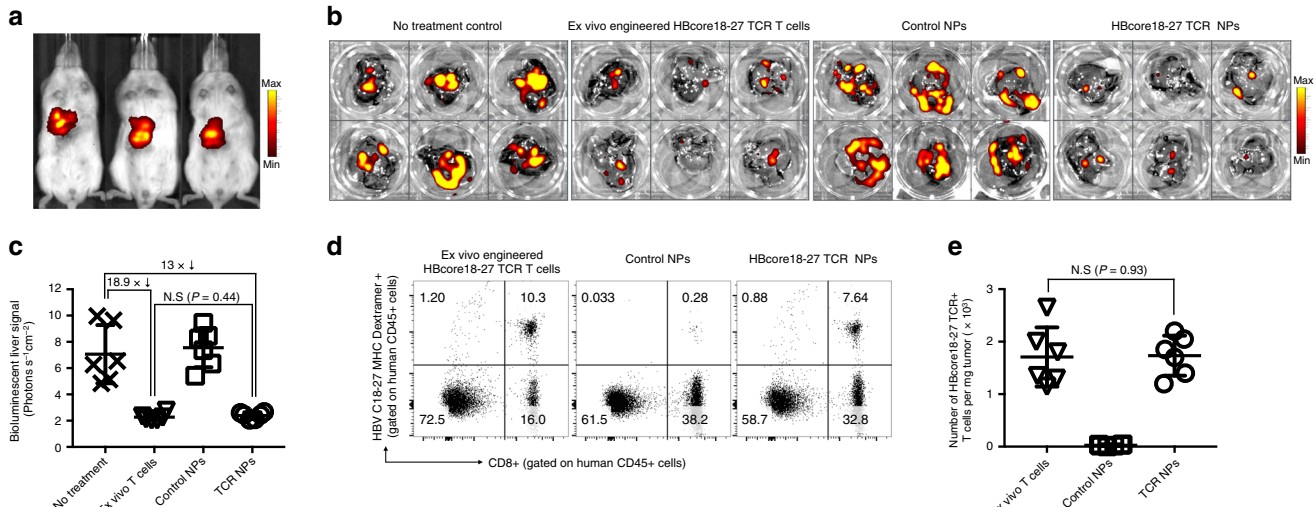

**Fig. 8 In situ programming of HBV-specific T cells using nanoparticles loaded with TCR transgenes. a** We established a mouse xenograft tumor model of HBV-induced HCC. HepG2 cells stably transduced with HBcAg and luciferase were surgically injected into the liver of NSG mice reconstituted with human T cells. Three weeks post-implantation, HepG2 tumors were visualized by in vivo bioluminescent imaging and assigned to nanoparticle (6 weekly doses of 50 μg mRNA encoding the HBcore18-17 TCR) or T-cell treatment (5 × 10⁶ T cells transduced ex vivo with lentiviral vectors encoding the HBcore18-17 TCR) groups. **b, c** Quantification of bioluminescent liver signal 6 weeks after treatment start. Shown are mean values ± SD; two tailed unpaired Student's *t*-test. N = 5 biologically independent samples. **d** Multicolor flow cytometry of cells recovered from the liver 18 days after treatment start. Adoptively transferred or in situ-programmed HBcore18-27 TCR+ T cells were identified by positive labeling for CD45, CD8, and MHC Pentamer. Absolute numbers are shown in **e**. Total cell counts of viable (trypan blue-negative) cells were multiplied by the percentage that was HBcore18–27 TCR+, CD8+, and CD45+. Shown are mean values ± SD. Pairwise differences in absolute numbers of T cells between the groups were statistically analyzed by two tailed unpaired Student's *t*-test. *P* < 0.05 was considered significant. N.s. nonsignificant. N = 5 biologically independent samples.

to ensure that changes in the parameters we measured could be attributed to the nanoparticles and not their reprogramming activity. Controls were either infused with 25 mM sodium acetate buffer (vehicle control) or received no injection. Animals were euthanized after 48 h, blood was collected to measure clinical biochemistry parameters, and we also performed a complete gross necropsy. The following tissues were evaluated by a board-certified staff pathologist: lung, liver, heart, brain, kidney, spleen, bone marrow, and duodenum. There were no histologic lesions that could be attributed to nanoparticle drug treatment (Fig. 10a). The few lesions noted were minimal to mild and considered incidental and unrelated to the study. All groups had two of five rats with minimal inflammatory infiltrates in the liver. These minimal infiltrates were considered to be background lesions and unrelated to treatment. Likewise, all groups contained individuals with minimal to mild chronic inflammation in the renal pelvis. The chronicity of the lesions was not consistent with an acute treatment effect, and therefore considered incidental. The complete blood count platelet values were slightly lower in the nanoparticle-treated animals relative to controls (mean 407 ± 115 K/μL vs. 290.4 ± 56.3 K/μL, respectively; *P* = 0.34, N.s.; Fig. 10b). Nanoparticle-treated animals were also mildly hypoglycemic relative to controls (mean 45.4 ± 26 mg/dL vs. 78.3 ± 69.8 mg/dL, respectively; P = 0.42, N.s.; Fig. 10c). All other serum chemistry values (including liver-function and kidney-function) of nanoparticle-treated rats were comparable to that of controls, indicating that systemic toxicities did not occur. Serum levels of interleukin-6 (IL-6) were moderately raised to an average 16.5 ± 5.9 pg/mL (Fig. 10d), which can be considered safe, based on previous report[44,45].

## Discussion

While CAR- and TCR-modified T cells have been transformational for a handful of blood cancers, it is clear that their current clinical applications represent only a sliver of the spectrum of possibilities that this technology might offer. In theory, both malignancies and chronic infections with targetable antigens could be treated with therapeutic T cells, as demonstrated in a large number of preclinical reports[20,22,46–49]. However, current methods to generate disease-specific T cells in vitro are elaborate and cannot support the treatment of sizable patient populations, because a new lymphocyte cohort must be produced for each patient. In an effort to make T-cell products more accessible to patients, the field has turned to allogeneic technologies to provide better scale and lower costs[50–52]. Several clinical-stage T-cell companies have begun testing CAR T cells manufactured as "off-the shelf" products from healthy unrelated donors, rather than the patient. While this approach allows treatment of cancer patients for whom autologous T cells cannot be manufactured due to the patient's low lymphocyte counts or poor T-cell quality, it requires several additional cell-engineering steps to prevent donor cells from attacking the host, and conversely to prevent a patient's own T cells from rejecting the infused product. This is most often accomplished using multiplex gene editing to eliminate native TCR and HLA molecules from the T-cell product[53]. But these additional manipulations add complexity, time and expense to the manufacturing process while reducing cell yield and viability[54]. To keep rejection at bay, patients receiving universal CAR T-cells are first heavily immunosuppressed through lymphodepleting chemotherapy, which requires time and exposes the patients to additional toxicity. It is therefore unlikely that ex vivo-engineered allogeneic cell products will dramatically expand the number of patients treated with T cells, especially those with infectious diseases which require swift interventions that leave the endogenous immune system intact.

Our group previously described an injectable DNA-based nanoreagent that can program circulating T cells with leukemia-specific CAR transgenes[55]. To overcome the inherently low gene transfer of plasmid DNA, which must enter the cell nucleus to be

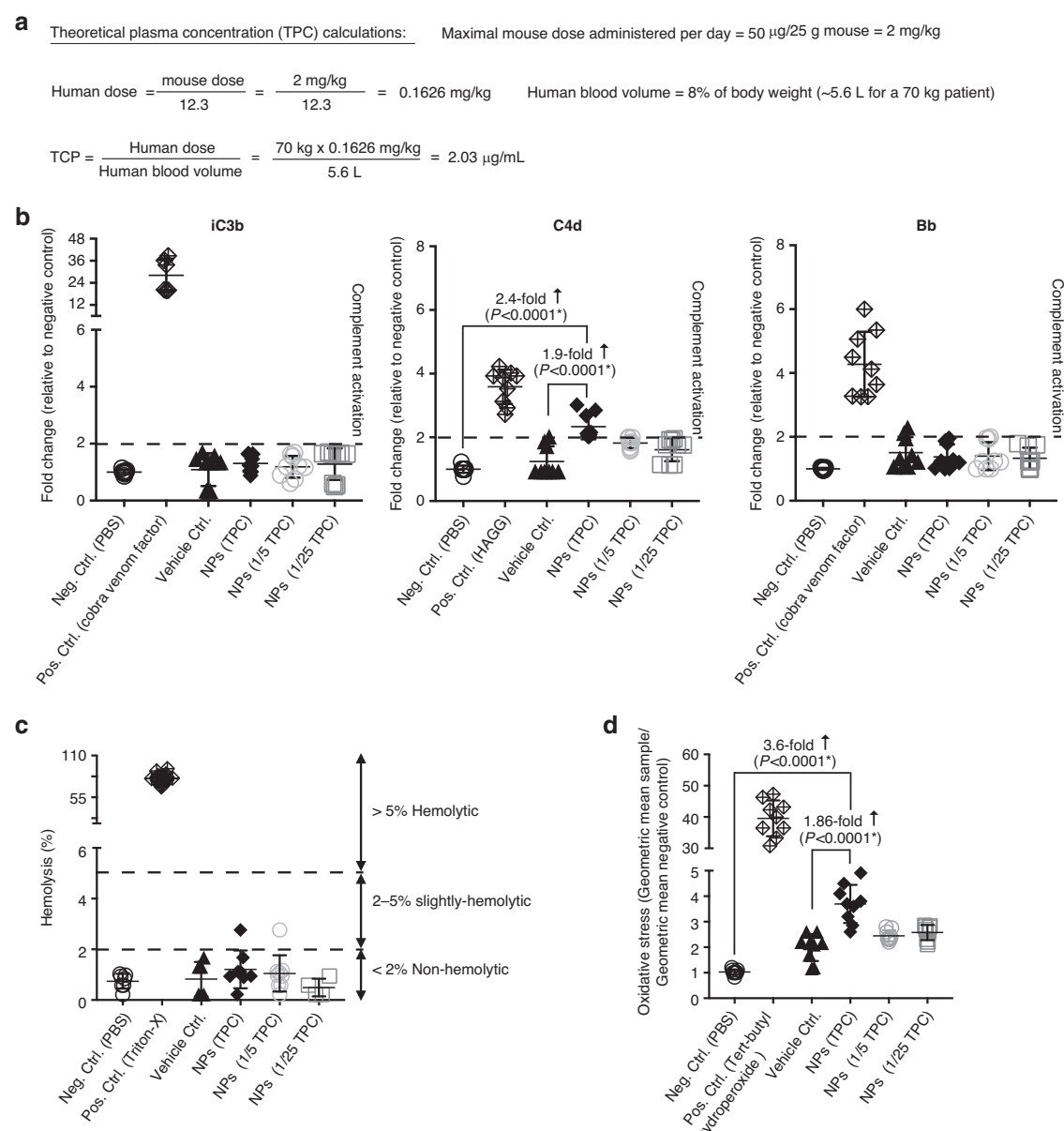

**Fig. 9 In vitro analysis of possible infusion reactions. a** Calculation of the theoretical plasma concentration. **b** Hemolytic activity of T-cell-targeted mRNA nanoparticles (NPs). **c** Quantitative determination of complement activation by an Enzyme Immunoassay. A 2-fold change relative to the negative PBS control was defined as the assay threshold (dashed line). **d** Mitochondrial oxidative stress in lymphocytes following NP transfection. In all panels of this figure, $N = 9$ biologically independent blood samples were analyzed. Also shown are the mean values ± SD. Pairwise differences between groups were analyzed by two tailed unpaired Student's *t*-test.

transcribed into mRNA, we loaded nanoparticles with a transposon/transposase system encoding the CAR, which randomly inserted into target cells' genomes. While that study provided proof-of-concept that in situ programming of CAR T cells using an injectable nanoreagent is possible, translation of this DNA nanomedicine into the clinic would have been challenging for the following reasons: (i) Unpredictable genotoxicity and expression kinetics. These nanoparticles stably integrate their therapeutic CAR transgenes into target cells, resulting in permanent genomic alterations and unpredictable genotoxicity in a variety of cell types. Furthermore, once nanoparticles are infused into patients, the physician has no control over the kinetics of in vivo CAR expression; (ii) Low copy numbers of relevant CAR genes per nanoparticle. The number of CAR genes we could load into these DNA nanoparticles was limited by the large size of the backbone

and promoter sequences of the plasmids as well as the requirement to co-deliver a transposase expression vector for stable integration. This substantially limits in situ gene transfer efficiencies, especially when trying to deliver large transgenes encoding TCR alpha and beta chains; and (iii) The need for abundant tumor antigen to expand the small population of in situ-transfected CAR T cells to therapeutically relevant numbers. This expansion period takes time, which is a disadvantage in patients with rapidly progressive disease or defined solid tumors.

Here, we explored the use of IVT mRNA to quickly and specifically program antigen-recognizing capabilities into circulating T cells as a strategy to treat cancer and infectious disease. In contrast to DNA nanocarriers, synthetic mRNA molecules are directly translated into therapeutic target proteins without the need to enter the nucleus, ensuring high transfection rates and

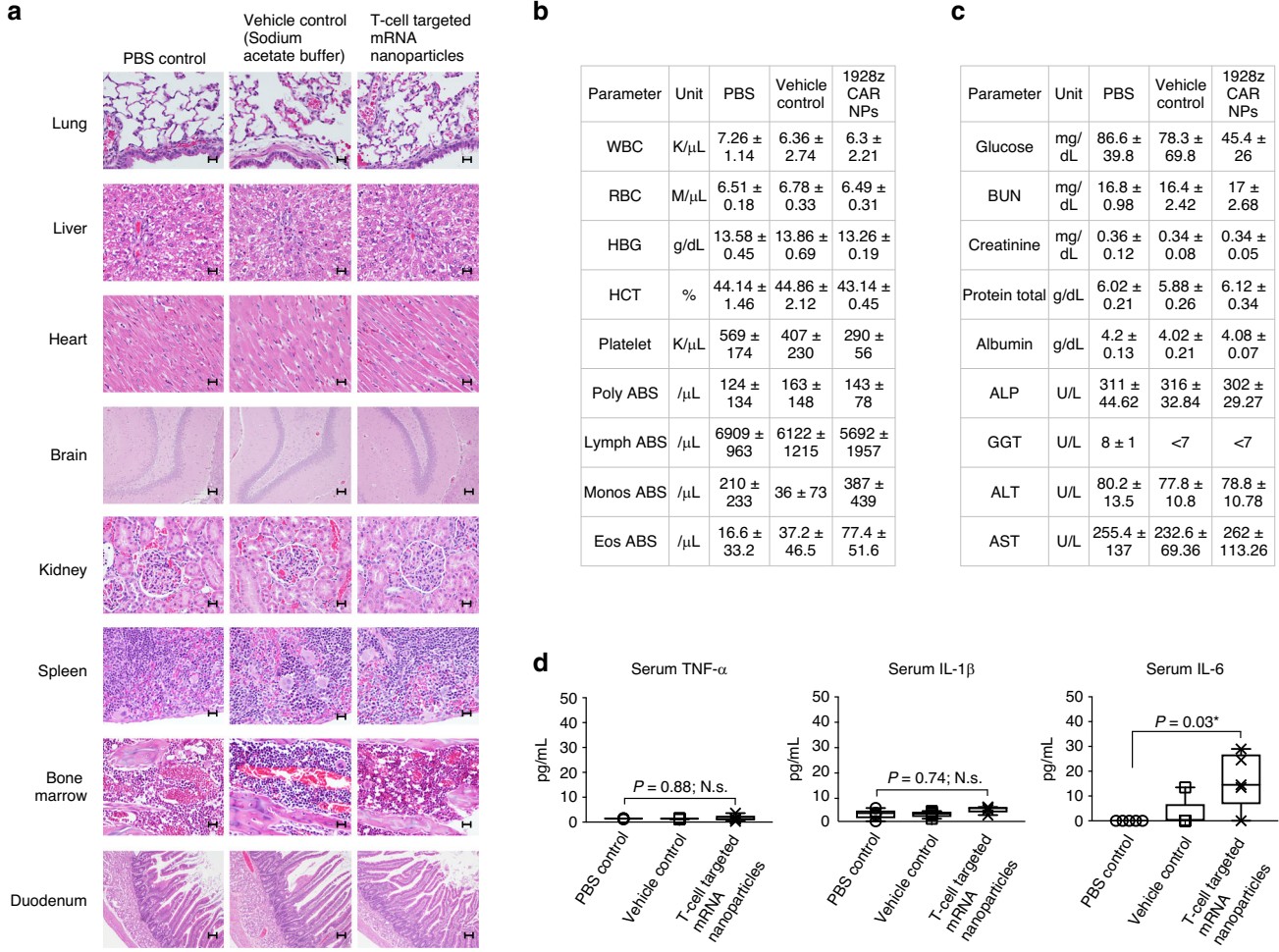

**Fig. 10 Infusions of nanocarriers are not associated with acute systemic toxicities.** Female Sprague Dawley rats were intravenously infused with CD8-targeted IVT mRNA encoding the 1928z CAR, and a full histopathological evaluation as well as serum chemistry analysis were performed 48 h later in a blinded fashion by a board-certified pathologist. **a** Representative H&E-stained sections of various organs isolated from controls or nanoparticle-treated animals. Scale bars, 400 µm. **b** Blood counts and **c** serum chemistry. **d** ELISA measurements of serum TNF-α, IL-1β, and IL-6 cytokines. On each box plot, the central mark indicates the median, and the bottom and top edges of the box indicate the interquartile range. Whiskers represent 95% confidence intervals. Two tailed unpaired Student's $t$-test. $N = 5$ biologically independent animals/group pooled from two independent experiments.

rapid therapeutic effects. Their trim size (in our study, the actual CAR-coding or TCR-coding sequence + only 276 bases for the 5′ UTR and polyA tract) results in a high copy number per nanoparticle. Also, uncontrolled insertional mutation and promoter dependency is avoided because the delivered mRNA exerts its function in the cytoplasm. We demonstrated that simple injection of rationally designed mRNA nanocarriers can selectively deliver CAR-genes or TCR-genes into host T cells and program them in quantities that are sufficient to bring about disease regression with efficacies that are similar to adoptive methods. Several ongoing clinical trials are testing repeated infusions of ex vivo-engineered mRNA CAR T cells in cancer patients (ClinicalTrials.gov: NCT01355965, NCT01897415, NCT02277522 and NCT02624258), and the first data suggest that transient CAR expression after cell infusion is sufficient to trigger antitumor responses[56–58].

Three important reasons why IVT mRNA has rapidly emerged as a new tool for adoptive T-cell therapy are its inherent safety, its highly efficient recombinant protein translation, and the ability to control pharmacokinetic properties of the therapy, similar to a conventional small molecule drug. Indeed, the kinetics of CAR-expressing T cells in mice that we measured following multiple dosing resemble the profile of a drug with a well-defined half-life (Fig. 4e). This is in sharp contrast to the rather unpredictable T-cell kinetics following adoptive transfer of engineered T cells, where the concentration of cells rises to a maximum in blood followed by a decline over a variable period of days to several months[59–62]. While in situ programming gains the ability to control pharmacokinetic properties of the therapy and to periodically reprogram fresh populations of host lymphocytes—thus potentially bypassing some of the major barriers to wider deployment of T-cell therapies (e.g., T-cell exhaustion and dysfunction, and also long-term toxicities)—the technology still has some limitations: (1) It relies on the presence of a sufficient number of functional T cells in the patients. Lymphopenia is frequent in patients with advanced cancers who have been heavily pretreated with chemotherapy agents. It is therefore likely that patient blood will need to be pre-screened ahead of enrollment into clinical trials of our off-the-shelf nanoreagent. (2) The efficacy of the drug could be blunted by elicited immune responses. Since T-cell-programming nanoparticles are administered periodically into patients with intact immunity, antidrug antibodies could form. For clinical translation of this technology, it will be important to choose fully humanized CD8-targeting ligands, deliver CAR/TCR constructs that carry a low immunological risk, and synthesize mRNA with pseudouridine (or the recently

described N1-methyl-pseudouridine[63]) and 5-methylcytosine to reduce innate immune responses.

To redirect circulating T cells to resident tumor cells in situ, several biotech drugmakers have developed bispecific antibodies, including BiTEs, DARTs, and diabodies[64–66]. Among these, blinatumomab (a CD19-specific BiTE) has shown encouraging results in clinical studies for patients with hematological malignancies[67–69]. However, BiTEs must be administered as a continuous infusion, which can produce systemic toxicities[67]. Furthermore, like conventional monoclonal antibodies, BiTEs do not undergo active biodistribution or self-amplification following infusion. By contrast, the nanoparticle-based gene-modification system described here can generate de novo tumor-specific T cells, which as a "living drug" actively localize to the target, increase in number, and serially destroy cancer cells. Interest in CAR T-cell therapy remains strong, with more than 300 ongoing CAR T-cell trials worldwide[70], so we believe it is timely to launch our off-the-shelf nanoreagent as a competing technology, that can quickly reprogram T cells to recognize and destroy tumors without the need for laboratory manipulations.

Prior to clinical studies in humans, we will extend our collaboration with the NCL to confirm safety of nanoparticles in a large-animal species, with reference to FDA regulations for nanomedicine. In contrast to therapeutics that are already established in the clinic, such as small molecules or antibodies, CAR-programming nanoparticles are multicomponent three-dimensional constructs that require a reproducible manufacturing process to reliably achieve the intended physicochemical characteristics, biological behaviors, and pharmacological profiles. The safety and efficacy of such nanomedicines can be influenced by minor variations in many parameters and need to be carefully monitored, particularly in the context of targeting to unintended sites and potential toxicities. Furthermore, nanomedicines require additional developmental and regulatory considerations compared with conventional medicines. Only a few facilities with the requisite degree of expertize are currently operational in the United States.

## Methods

**Cell lines.** Raji lymphoma cells were obtained from ATCC (Cat# CCL86) and cultured in RPMI 1640 containing 10% fetal bovine serum (FBS), 0.8 mM L-glutamine, 25 mM HEPES buffer, and 1% penicillin–streptomycin. The human prostate adenocarcinoma cell line LNCaP C42 was provided by Dr. Michel Sadelain (Memorial Sloan-Kettering Cancer Center). These cells were cultured in RPMI 1640 supplemented with 10% FBS, 2 mM L-glutamine, 1.0 mM sodium pyruvate, and 1% penicillin–streptomycin. The human hepatoma cell line HepG2-core, stably transduced with HBcAg, was provided by Dr. Antonio Bertoletti (Duke-NUS Medical School). The base medium for this cell line is Eagle's Minimum Essential Medium (ATCC Cat# 30-2003). The Eμ-ALL01 cell line (a gift from Dr. Michel Sadelain; Memorial Sloan-Kettering Cancer Center, New York, NY)[25] was cultured in complete RPMI 1640 medium with 10% heat-inactivated fetal bovine serum (FBS), 2 mM L-glutamine, 1.5 g/L sodium bicarbonate, 4.5 g/L glucose, 10 mM HEPES, 1.0 mM sodium pyruvate, and 0.05 mM 2-mercaptoethanol. For bioluminescent imaging, all cell lines were retrovirally transduced with firefly luciferase (F-luc). All cell lines tested negative for mycoplasma using a DNA-based PCR test (DDC Medical).

**Retroviral vectors and virus production.** A retroviral SFG-vector expressing the anti-CD19-28z CAR was kindly provided by Dr. Michel Sadelain (Memorial Sloan-Kettering Cancer Center, New York). The anti-ROR1-28z CAR was custom-designed by Creative Biolabs (Shirley, NY). It consists of a single-chain antibody targeting human ROR1 linked via a c-myc tag to a synthetic receptor skeleton comprised of the CD8 hinge, the CD28 transmembrane and signaling domains, and the signaling domain from CD3ζ. This construct was then cloned by VectorBuilder (Santa Clara, CA) into their MMLV Retrovirus Gene Expression Vector. A MMLV retroviral vector containing the Vα and Vβ chains of the HBcore18–27-specific TCR sequences[71] was cloned by VectorBuilder (vectorbuilder.com). To generate retroviral particles, we transfected the Plat-A retroviral packaging cell line (cellbiolabs.com) according to the manufacturer's instructions and harvested retroviral supernatant 48 h later.

**Retroviral transduction of primary human T cells.** To generate CAR-transduced or TCR-transduced T cells, human CD8+ T cells (obtained from the Hematopoietic Cell Processing and Repository Core at the Fred Hutchinson Cancer Research Center) were stimulated with antiCD3/CD28-coated Dynabeads (ThermoFisher Scientific) for 48 h at a bead-to-cell ratio of 1:1 in the presence of 30 U/mL rIL-2. Beads were then magnetically removed and transferred to retronectin-coated plates (Takara) with virus, then spin-infected for 1 h at 3000×g at 32 °C. Following a second spinoculation in retroviral supernatant the next day, the cells were cultured for 24 h in the presence of IL-2 before using them as a tumor therapeutic.

**PbAE synthesis.** We combined 1,4-butanediol diacrylate with 4-amino-1-butanol in a 1:1 molar ratio of diacrylate to amine monomers. Acrylate-terminated poly(4-amino-1-butanol-co-1,4-butanediol diacrylate) was formed by heating the mixture to 90 °C with stirring for 24 h. 2.3 g of this polymer was dissolved in 2 mL tetrahydrofuran (THF). To form the piperazine-capped 447 polymer, 786 mg of 1-(3-aminopropyl)-4-methylpiperazine in 13 mL THF was added to the polymer/THF solution and stirred at room temperature (RT) for 2 h. The capped polymer was precipitated with five volumes of diethyl ether, washed with two volumes of fresh ether, and dried under vacuum for 1 day. Neat polymer was dissolved in dimethyl sulfoxide (DMSO) to a concentration of 100 mg/mL and stored at −20 °C.

**Preparation of CD8-targeting antibodies.** Antihuman CD8α antibody (clone OKT-8) was purchased from BioXcell (Cat# #BE0004-2). Before use, Fc-glycans were deglycosylated using deGlycIT™ spin columns containing IgGZERO enzyme (Genovis, Cat# #A0-IZ6-10), according to manufacturer instructions.

**AntiCD3 murine IgG2a LALA-PG and nonbinder control (used only for in vivo experiments in immunocompetent mice/Fig. 6).** The antiCD3 scFv design was based on the bispecific antibody construct Epcam x 2C11 scFv (GenBank: SJL88240.1). Detailed sequence information can be found in the Supplementary Info PDF.

**Antibody conjugation to PGA.** Fifteen kilodalton of poly-glutamic acid (from Alamanda Polymers, Cat# PLE100) was dissolved in water to form 20 mg/mL and sonicated for 10 min. An equal volume of 4 mg/mL 1-ethyl-3-(3-dimethylamino-propyl) carbodiimide hydrochloride (Thermo Fisher) in water was added, and the solution was mixed for 5 min at RT. The resulting activated PGA was then combined with antibodies at a 4:1 molar ratio in phosphate buffered saline (PBS) and mixed for 4 h at RT. To remove unlinked PGA, the solution was diafiltrated through Amicon Ultra Centrifugal Filters (50 K MWCO). Antibody concentrations were determined by absorbance at 280 nm.

**mRNA synthesis.** Codon-optimized mRNA for eGFP, the antiCD19-28z CAR[25], the antiROR1-28z CAR (Creative Biolabs Cat# CAR-T-1-M324-2Z), and the HBcore18-27-specific TCR[72] were manufactured by TriLink Biotechnologies and were capped with the anti-reverse Cap Analog 3′-O-Me-m7G(5′)ppp(5′)G (ARCA), and fully substituted with the modified ribonucleotides pseudouridine (Ψ) and 5-methylcytidine (m5C).

**Nanoparticle preparation.** mRNA stocks were diluted to 100 μg/mL in 25 mM nuclease-free sodium acetate buffer, pH 5.2 (NaOAc). PBAE-447 polymer in DMSO was diluted to 6 mg/mL in NaOAc, and added to mRNA at a 60:1 (w:w) ratio. After the resulting mixture was vortexed for 15 s at medium speed, it was incubated for 5 min at RT so nanoparticles (NPs) could form. To add targeting elements to the NPs, PGA-linked antibodies were diluted to 250 μg/mL in NaOAc and added at a 2.5:1 (w:w) ratio to the mRNA. The resulting mixture was vortexed for 15 s at medium speed, and then incubated for 5 min at RT to permit binding of PGA-Ab to the NPs. The NPs were lyophilized by mixing them with 60 mg/ml D-sucrose as a cryoprotectant, and flash-freezing them in liquid nitrogen, before processing them in a FreeZone 2.5 L Freeze Dry System (Labconco). The lyophilized NPs were stored at −80 °C until use. For application, lyophilized NPs were re-suspended in a volume of sterile water to restore their original concentration.

**Characterization of nanoparticle size distribution, concentration, ζ-potential, and mRNA encapsulation.** The physicochemical properties of NPs (including hydrodynamic radius, polydispersity, ζ-potential, and stability) were characterized using a Zetapals instrument (Brookhaven Instrument Corporation) at 25 °C. To measure the hydrodynamic radius and polydispersity based on dynamic light scattering, NPs were diluted 5-fold in 25 mM NaOAc (pH 5.2). To measure the ζ-potential, NPs were diluted 10-fold in 10 mM PBS (pH 7.0). To assess the stability and concentration of NPs, freshly prepared particles were diluted in 10 mM PBS buffer (pH 7.4). The hydrodynamic radius and polydispersity of NPs were measured every 10 min for 5 h, and their sizes and particle concentrations were derived from Particle Tracking Analysis using a Nanosite 300 instrument (Malvern). A Qubit RNA HS assay kit (ThermoFisher, Cat# Q32852) was used for mRNA quantification. It contains a proprietary cyanine dye that specifically binds to the nucleic acid. NP samples were diluted 20-fold in 385 mM HFIP+ 14.5 mM TEA in

5% (v/v) MeOH. Calibration mRNA stock sample was prepared and mixed at a ratio of IRF5:IKKb = 25:8 with a final concentration at 8 µg/mL in 385 mM HFIP+ 14.5 mM TEA in 5% (v/v) MeOH. Calibration standards were prepared by diluting stock in 385 mM HFIP+ 14.5 mM TEA in 5% (v/v) MeOH accordingly (0.2–4.0 µg/mL range). Both NP samples and standard samples were incubated at 60 °C for 60 min. Each sample (10 µL) was mixed immediately with 190 µL of assay working solution and vortexed for 2–3 s, then incubated at RT for 2 min before measuring.

**Nanoparticle transfection of ex vivo-cultured T cells.** Cryopreserved PBMCs from normal donors were thawed by drop-wise addition of warm T-cell culture medium (TCM: RPMI 1640, supplemented with 10% FBS, 2mM L-glutamine, 1.0 mM sodium pyruvate, 1% penicillin–streptomycin, and 0.05 mM ß-mercaptoethanol), followed by centrifugation. CD8 T cells were isolated by negative selection (Stemcell). Cells were cultured in TCM+IL-2 at 10^6 cells/mL and stimulated with CD3/CD28 beads (Dynabeads, Life Technologies) at a 1:1 bead:cell ratio for 24 h. These beads were then removed and cells were rested for another 24 h before NP-transfection. For NP-mediated transfections, the T cells were resuspended in Stemcell ImmunoCult™-XF T Cell Expansion Medium at a concentration of 1 × 10^6/mL. Antibody-targeted NPs containing 3 µg of mRNA/10^6 cells were mixed into this suspension for an exposure of 2 h at 37 °C before diluting it four times by adding T-cell medium supplemented with 50 IU IL-2/mL.

**Flow cytometry.** Data were acquired using a BD LSRFortessa or FacsCanto II cell analyzer running FACSDIVA software, and analyzed with FlowJo v10.1. Antibodies and other staining reagents used in flow cytometry are summarized below. Data were collected using a BD LSRFortessa analyzer running FACSDIVA software (Beckton Dickinson).

Human CD8 (clone 3B5, dilution 1:400, APC-Alexa Fluor 750, ThermoFisher Cat#MHCD0827).
Human CD3 (clone UCHT1, dilution 1:400, Alexa Fluor 532, ThermoFisher Cat#58-0038-42).
Human CD45 (clone HI130, dilution 1:400, PE-Cyanine 7, ThermoFisher Cat#25-0459-42).
C-Myc (clone 9B11, dilution 1:50, Cellsignal.com Cat# 3739s).
Human ROR1 (dilution 1:200, Antibodies-online.com Cat#ABIN4899817).
PSMA (dilution 1:300, FITC, LifeSpan BioSciences Cat# LS-C317473-50).
HLA-A201-HBV core 18–27 pentamer (dilution 1:200, PE, Proimmune Cat# F023-2B-23-A*02:01-FLPSDFFPSV-Pentamer0150 test R-PE).
Live/Dead Fixable Green (dilution 1:800, FITC, Life Technologies Cat# L23101).
Mouse CD45 (clone 30-F11, dilution 1:100, BUV661, BD Biosciences Cat# 612975).
Mouse CD11b (clone M1/70, dilution 1:100, BUV395, BD Biosciences Cat# 563553).
Mouse CD11c (clone HL3, dilution 1:100, APC-Cy™7, BD Biosciences Cat# 561241).
Mouse LY6G (clone 1A8, dilution 1:100, BUV737, BD Biosciences Cat# 741813).
Mouse CD19 (clone 1D3, dilution 1:100, BUV805, BD Biosciences Cat# 749027).
Mouse F4/80 (clone BM8, dilution 1:200, FITC, Biolegend Cat# 123108)
Mouse CD4 (clone RM4-5, dilution 1:100, Brilliant Violet 421, Biolegend Cat# 100563)
Mouse CD8a (clone 53-6.7, dilution 1:50, PerCP/Cyanine5.5, Biolegend Cat# 100734)
Mouse CD44 (clone IM7, dilution 1:100, BUV496, BD Biosciences Cat# 741057)
Mouse CD62L (clone MEL-14, dilution 1:50, BV650, Biolegend Cat# 104453)
Mouse CD69 (clone H1.2F3, dilution 1:100, BUV563, BD Biosciences Cat# 741234)
Live/dead stain—Zombie Aqua (dilution 1:400, BV510, Biolegend Cat# 423101)
Annexin A5 (dilution 1:20, APC/Fire™ 750, Biolegend Cat# 640953)
7-AAD (dilution 1:100, Biolegend Cat# 420404)

**Real-time tumor cell killing assay.** Raji cells and HepG2-core tumor cells were stably transduced using red fluorescent protein NucLight Red lentiviral reagent (Essen Bioscience). NucLight Red target cells were co-cultured in 96-well plates with retrovirus-transduced or nanoparticle-transfected 1928z CAR T cells (Fig. 3d), or HBcore-18-27 TCR T cells (Fig. 3i) for 45 h in TCM medium. The normalized percentage of live tumor cells was calculated from the number of NucLight Red target cells acquired using IncuCyte Zoom software, at the noted time points (every 15 min), normalized against targets alone and against 0 h in coculture groups. A normalized killing count graph was generated using GraphPad Prism software.

**Cytokine secretion assays.** T-cell cytokine release was measured with ELISA (R&D Systems) 24 h (IL-2) or 48 h (IFN-γ and TNF-α) after stimulation on irradiated Raji lymphoma cells (Fig. 3e) or HepG2-HBcAg hepatocellular carcinoma cells (Fig. 3j).

**Nanoparticle characterization laboratory assay cascade protocols.** The nanoparticle in vitro toxicity assays were conducted according to standardized analytical cascade assays published on the NCL webpage: https://ncl.cancer.gov/resources/assay-cascade-protocols.

Hemolysis was analyzed by determining the plasma-free hemoglobin, which is indicative of erythrocyte damage by nanoparticles as per the NCL Method ITA-1. Briefly, whole blood was collected in K2-EDTA-coated tubes from three different donors, and the plasma-free hemoglobin was calculated to determine the suitability of the donor blood samples for the hemolysis assay. PBAE-C18V-α-β mRNA nanoparticles were freshly prepared as described above. Theoretical Plasma Concentration (TPC) for the nanoparticles was calculated as follows:
Max mouse dose administered per day = 50 µg/25 g mouse = 2 mg/kg.
Human dose = mouse dose/12.3 = 2/12.3 = 0.1626 mg/kg.
Blood volume for 70 kg human = 5.6 L (8% of body weight)
TPC = Human dose/blood volume = (70 kg * 0.1626 mg/kg) / 5.6 L = 2.0325 µg/mL.

Freshly prepared PBAE nanoparticles were incubated with blood at 2× TPC, TPC, TPC/5, TPC/25 for 3 h at 37 °C with constant mixing. Post 3 h, the tubes were centrifuged at 800×g for 15 min and the supernatant was used to determine the plasma-free hemoglobin, due to nanoparticle treatment, using cyanmethemoglobin reagent and hemoglobin standards. Absorbance was read at 540 nm. One percent of Triton-X was used as positive control, 1× phosphate buffer was used as negative control and 25 mM sodium acetate was used as vehicle control. Percent hemolysis was determined as follows.
% Hemolysis = (hemoglobin in test sample/total blood hemoglobin) * 100.

Complement activation was analyzed by determining the levels of C4d (indicative of complement activation via classical or lectin pathway), Bb (indicative of complement activation via alternative pathway) and iC3b (indicative of C3 component of complement activation pathways) according to ITA-5.2. Briefly, fresh plasma was isolated from whole blood from three different donors. PBAE-C18V-α-β mRNA nanoparticles were freshly prepared as described above. PBAE nanoparticles were incubated in equal volumes (100 µL each) of veronal buffer and plasma at 5× TPC, TPC, TPC/5, TPC/25 for 30 min at 37 °C after votexing. Hundred microliter of aliquots were frozen at −20 °C to determine the levels of C4d, Bb and iC3b using Micro Vue (Quidel Corp.) EIA kits (A008, A027, A006, respectively) as per the manufacturer's instructions. Cobra venom factor was used as a positive control for iC3b and Bb levels, and HAGG/Doxil was used as a positive control for C4d levels. 1× phosphate buffer was used as a negative control and 25 mM sodium acetate was used as a vehicle control. The C4d, Bb and iC3b levels were plotted as fold change compared to the negative control.

Oxidative stress in T cells was analyzed using MitoSox reagent according to ITA-32. Briefly, 800 µL of TALL-104 cell suspension at 1 × 10^6 cells/mL was added to a 24-well plate. PBAE-C18V-α-β mRNA nanoparticles were prepared as described above and lyophilized with 60 mg/mL sucrose solution. Two hundred microliter of PBAE nanoparticles were incubated with 800 µL of cell suspension at 5× TPC, TPC, TPC/5, TPC/25 for 30 min, 3 and 6 h in a 37 °C, 5% CO_2 incubator. Post incubation, cells were washed with 1× PBS and incubated with MitoSox reagent (5 µM final concentration) for 30 min at RT. Cells were then washed thrice with 1× PBS and analyzed by flow cytometry using the PE channel. Tert-butyl-hydroperoxide was used as positive control, 1× phosphate buffer was used as negative control, and 25 mM sodium acetate in 60 mg/mL sucrose solution lyophilized and resuspended in distilled water was used as a vehicle control. All samples were analyzed in triplicates. Data were plotted as the geometric mean of the sample divided by the geometric mean of the negative control.

**Mice and in vivo tumor models.** Animals were housed in the rodent barrier facility of Fred Hutchinson Cancer Research Center, and used in the context of an animal protocol approved by the Center's Institutional Animal Care and Use Committee (Protocol ID: PROTO000050782, Protocol Title: 50782, Adoptive Cell Therapy bioengineering). A 14-h light/10-h dark cycle and temperatures of 65–75 °F (~18–23 °C) with 40–60% humidity were used. Experimental and control animals were co-housed.

Raji tumor model: NOD/SCID/IL2r−/− (NSG) mice were bred in house and engrafted via tail vein with 1 × 10^6 Raji cells expressing firefly luciferase. Once disease established, mice were carefully monitored for signs of progressive hindlimb paralysis and body weight loss as a consequence of tumor spread in the bone marrow. Tumors were serially monitored using bioluminescence imaging, and animals exceeding a whole-animal bioluminescent tumor photon count of >20 × 10^6 s^{-1} cm^{-2} sr^{-1} (defined as experimental endpoint) were euthanized by CO_2.

Eµ-ALL01 leukemia model: Eµ-ALL01 leukemia cells[25] were injected into the tail vein of 4–6 week-old female albino B6 (C57BL/6J-Tyr<c-2J>) mice (Jackson Laboratories, Stock No: 000058) and allowed to develop for five days. Following tail vein injection of Eµ-ALL01 leukemia cells, the mice were randomly assigned to experimental cohorts. Mice were serially monitored using bioluminescent imaging to closely monitor tumor progression. We defined a whole-animal bioluminescent tumor photon count of >20 × 10^6 s^{-1} cm^{-2} sr^{-1} as experimental endpoint, at which point animals were euthanized by CO_2. Animals that did not reach this endpoint were euthanized by CO_2 at day 54.

Orthotopic LNCaP C42 prostate tumor model: For orthotopic prostate tumor cell implantation, a longitudinal incision was made in the lower abdomen of anesthetized male NSG mice. The bladder, seminal vesicles, and prostate were partially extravasated from the abdominal cavity to expose the dorsal lobes of the prostate. Using a 27-gauge needle, 2 × 10^6 LNCaP C42 tumor cells were injected into each lobe in 25 µL of PBS. The abdominal cavity, muscle and skin were closed in two layers

with Vicryl Rapide polyglactin 910 P-1 11 mm absorbable sutures and 7 mm stainless steel wound clips, respectively. Tumors developed over the course of ~21 days post implantation, and were serially monitored using bioluminescent imaging. We defined a whole-animal bioluminescent tumor photon count of $>30 \times 10^6\ s^{-1}\ cm^{-2}\ sr^{-1}$ as experimental endpoint, at which point animals were euthanized by $CO_2$.

Mouse xenograft tumor model of HBV-induced hepatocellular carcinoma (HCC): HepG2 cells stably transduced with HBcAg and luciferase were surgically injected into the liver of NSG mice. Specifically, $10 \times 10^6$ tumor cells were resuspended in 250 μL of serum-free medium. Directly before injection, we mixed the cell suspension at a 1:1 (vol/vol) ratio with Matrigel to increase the viscosity of the injected cell suspension. Small 0.5-mL syringes with 28.5-gauge needles were used to inject the cell/Matrigel suspension. To avoid leakage of tumor cells from the injection site—which might lead to local spread and "seeding" metastasis in the peritoneal cavity—we limited the injected volume to 20–30 μL. A steady and slow injection was performed to prevent leakage of the injected cell suspension, and to minimize damage to the surrounding liver tissue. After removal of the needle, the liver surface at the site of the needle tract was covered with Gelfoam for 5 min to minimize bleeding and potential backflow/leakage. Nine weeks following tumor implantation was defined as experimental endpoint, at which point all animals were euthanized by $CO_2$ to isolate organs for analysis.

**In vivo bioluminescence imaging**. We used D-Luciferin (Xenogen) in PBS (15 mg/mL) as a substrate for F-luc (imaging of Raji-luc lymphoma cells, LNCap C42-luc, and HepG2-luc hepatocellular carcinoma cells). Bioluminescence images were collected with a Xenogen IVIS Spectrum Imaging System (Xenogen, Alameda, CA). Living Image software version 4.3.1 (Caliper Life Sciences) was used to acquire (and later quantitate) the data 10 min after intraperitoneal injection of D-luciferin into animals anesthetized with 150 mg/kg of 2% isoflurane (Forane, Baxter Healthcare). Acquisition times ranged from 10 s to 5 min.

**Toxicity analysis in rats**. To measure potential in vivo toxicities of infusing T cell-targeting mRNA nanoparticles, we injected female, 6–8-week-old Sprague-Dawley rats (5/group) intravenously with a single dose of CD8-targeted nanoparticles, carrying 100 μg mRNA encoding the 1928z CAR. Controls were either infused with 50 mM sodium acetate buffer (vehicle control) or received no injection. Forty-eight hours after the infusion, animals were anesthetized and blood was collected by cardiac bleed to determine the complete blood counts. Blood was also collected for serum chemistry and cytokine profile analyses (performed by Phoenix Central Laboratories, Mukilteo, WA). Animals were then euthanized with $CO_2$ to retrieve organs, which were washed with deionized water before fixation in 4% paraformaldehyde. The tissues were processed routinely, and sections were stained with hematoxylin and eosin. The specimens were interpreted by Dr. Amanda Koehne, DVM, DACVP, a board-certified staff pathologist, in a blinded fashion.

**Statistical analysis**. The statistical significance of observed differences was analyzed using the unpaired, two-tailed Student's $t$-test or the unpaired, two-tailed one-way ANOVA test. The $P$ values for each measurement are listed in the figures or figure legends. We characterized survival data using the Log-rank test. All statistical analyses were performed using either GraphPad Prism software version 7.0b or R software.

**Study approval**. Blood samples were obtained from healthy donors. Donors provided written informed consent for research protocols approved by the Institutional Review Board of the FHCRC. The care and use of mice in this study was approved by the Institutional Animal Care & Use Committee (Dr. George Georges, MD, IACUC chair) at the Fred Hutchinson Cancer Research Center and complied with all relevant ethical regulations for animal testing and research (Assurance #A3226-01, IACUC Protocol Number 50782).

**Reporting summary**. Further information on research design is available in the Nature Research Reporting Summary linked to this article.

## Data availability

All data supporting the findings of this study are available within the article and its supplementary information files and directly from M. Stephan upon reasonable request. A reporting summary for this article is available as a Supplementary Information file. Source data are provided with this paper.

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

# ARTICLE

34. Liang, T. J. Hepatitis B: the virus and disease. *Hepatology* **49**, S13–S21 (2009).
35. Szebeni, J., Simberg, D., Gonzalez-Fernandez, A., Barenholz, Y. & Dobrovolskaia, M. A. Roadmap and strategy for overcoming infusion reactions to nanomedicines. *Nat. Nanotechnol.* **13**, 1100–1108 (2018).
36. Dobrovolskaia, M. A. & McNeil, S. E. Understanding the correlation between in vitro and in vivo immunotoxicity tests for nanomedicines. *J. Control Release* **172**, 456–466 (2013).
37. Standard Test Method for Analysis of Hemolytic Properties of Nanoparticles. ASTM E2524 - 08(2013). American Society for Testing and Materials. https://statnano.com/standard/astm/50/ASTM-E2524-08 (2013).
38. Finkel, T. & Holbrook, N. J. Oxidants, oxidative stress and the biology of ageing. *Nature* **408**, 239–247 (2000).
39. Manke, A., Wang, L. & Rojanasakul, Y. Mechanisms of nanoparticle-induced oxidative stress and toxicity. *Biomed. Res. Int.* **2013**, 942916 (2013).
40. Janas, M. M. et al. The nonclinical safety profile of GalNAc-conjugated RNAi therapeutics in subacute studies. *Toxicol. Pathol.* **46**, 735–745 (2018).
41. Bouchard, P. Toxicological considerations for oligonucleotide therapeutics. http://www.toxicology.org/groups/rc/NorCal/docs/2010Spring/2010_3ToxConsider_OligonucleotideTherap.pdf (2010).
42. Sedic, M. et al. Safety evaluation of lipid nanoparticle-formulated modified mRNA in the sprague-dawley rat and cynomolgus monkey. *Vet. Pathol.* **55**, 341–354 (2018).
43. Nair, A. B. & Jacob, S. A simple practice guide for dose conversion between animals and human. *J. Basic Clin. Pharm.* **7**, 27–31 (2016).
44. Tarrant, J. M. Blood cytokines as biomarkers of in vivo toxicity in preclinical safety assessment: considerations for their use. *Toxicol. Sci.* **117**, 4–16 (2010).
45. Copeland, S. et al. Acute inflammatory response to endotoxin in mice and humans. *Clin. Diagn. Lab. Immunol.* **12**, 60–67 (2005).
46. Hale, M. et al. Engineering HIV-resistant, anti-HIV chimeric antigen receptor T cells. *Mol. Ther.* **25**, 570–579 (2017).
47. Parida, S. K. et al. T-Cell therapy: options for infectious diseases. *Clin. Infect. Dis.* **3**, S217–S224 (2015).
48. Kumaresan, P. R., da Silva, T. A. & Kontoyiannis, D. P. Methods of controlling invasive fungal infections using CD8(+) T cells. *Front. Immunol.* **8**, 1939 (2017).
49. Wisskirchen, K. et al. T cells grafted with HBV-specific T-cell receptors of high functional avidity achieve functional cure of HBV infection in humanized mice. *J. Hepatol.* **68**, S19–S19 (2018).
50. Qasim, W. et al. Preliminary results of UCART19, an allogeneic anti-CD19 CAR T-cell product in a first-in-human trial (PALL) in pediatric patients with CD19+relapsed/refractory B-cell acute lymphoblastic leukemia. *Blood* **130**, 887–887 (2017).
51. Sommer, C. et al. Preclinical evaluation of allogeneic CAR T cells targeting BCMA for the treatment of multiple myeloma. *Mol. Ther.* **27**, 1126–1138 (2019).
52. Qasim, W. Allogeneic CAR T cell therapies for leukemia. *Am. J. Hematol.* **94**, S50–S54 (2019).
53. Graham, C., Jozwik, A., Pepper, A. & Benjamin, R. Allogeneic CAR-T cells: more than ease of access? *Cells* **7**, 155 (2018).
54. Moffett, H. F. et al. Hit-and-run programming of therapeutic cytoreagents using mRNA nanocarriers. *Nat. Commun.* **8**, 389 (2017).
55. Smith, T. T. et al. In situ programming of leukaemia-specific T cells using synthetic DNA nanocarriers. *Nat. Nanotechnol.* **12**, 813–820 (2017).
56. Beatty, G. L. et al. Activity of mesothelin-specific chimeric antigen receptor T cells against pancreatic carcinoma metastases in a phase 1 trial. *Gastroenterology* **155**, 29–32 (2018).
57. Wang, C. M. et al. Autologous T cells expressing CD30 chimeric antigen receptors for relapsed or refractory Hodgkin lymphoma: an open-label phase I trial. *Clin. Cancer Res.* **23**, 1156–1166 (2017).
58. Foster, J. B., Barrett, D. M. & Kariko, K. The emerging role of in vitro-transcribed mRNA in adoptive T cell immunotherapy. *Mol. Ther.* **27**, 747–756 (2019).
59. Mailankody, S. et al. Clinical responses and pharmacokinetics of MCARH171, a human-derived Bcma targeted CAR T cell therapy in relapsed/refractory multiple myeloma: final results of a phase I clinical trial. *Blood* **132**, 959–959 (2018).
60. Locke, F. L. et al. Phase 1 results of ZUMA-1: a multicenter study of KTE-C19 anti-CD19 CAR T cell therapy in refractory aggressive lymphoma. *Mol. Ther.* **25**, 285–295 (2017).
61. Gardner, R. A. et al. Intent-to-treat leukemia remission by CD19 CAR T cells of defined formulation and dose in children and young adults. *Blood* **129**, 3322–3331 (2017).
62. Mueller, K. T. et al. Cellular kinetics of CTL019 in relapsed/refractory B-cell acute lymphoblastic leukemia and chronic lymphocytic leukemia. *Blood* **130**, 2317–2325 (2017).
63. Svitkin, Y. V. et al. N1-methyl-pseudouridine in mRNA enhances translation through eIF2alpha-dependent and independent mechanisms by increasing ribosome density. *Nucleic Acids Res.* **45**, 6023–6036 (2017).
64. Suryadevara, C. M. et al. Are BiTEs the "missing link" in cancer therapy? *Oncoimmunology* **4**, e1008339 (2015).
65. Sung, J. A. et al. Dual-affinity re-targeting proteins direct T cell-mediated cytolysis of latently HIV-infected cells. *J. Clin. Invest.* **125**, 4077–4090 (2015).
66. Fan, D. et al. Redirection of CD4+ and CD8+ T lymphocytes via an anti-CD3 x anti-CD19 bi-specific antibody combined with cytosine arabinoside and the efficient lysis of patient-derived B-ALL cells. *J. Hematol. Oncol.* **8**, 108 (2015).
67. Topp, M. S. et al. Safety and activity of blinatumomab for adult patients with relapsed or refractory B-precursor acute lymphoblastic leukaemia: a multicentre, single-arm, phase 2 study. *Lancet Oncol.* **16**, 57–66 (2015).
68. Keating, A. K. et al. Reducing minimal residual disease with blinatumomab prior to HCT for pediatric patients with acute lymphoblastic leukemia. *Blood Adv.* **3**, 1926–1929 (2019).
69. Curran, E. & Stock, W. Taking a "BiTE out of ALL": blinatumomab approval for MRD-positive ALL. *Blood* **133**, 1715–1719 (2019).
70. Han, X., Wang, Y. & Han, W. D. Chimeric antigen receptor modified T-cells for cancer treatment. *Chronic Dis. Transl. Med.* **4**, 225–243 (2018).
71. Adriani, G., Pavesi, A. & Kamm, R. D. Studying TCR T cell anti-tumor activity in a microfluidic intrahepatic tumor model. *Methods Cell Biol.* **146**, 199–214 (2018).
72. Pavesi, A. et al. A 3D microfluidic model for preclinical evaluation of TCR-engineered T cells against solid tumors. *JCI Insight* **2**, e89762 (2017).

## Acknowledgements

This work was supported in part by the FHCRC Immunotherapy Initiative with funds provided by the Bezos Family Foundation, by the National Science Foundation (CAREER Award #1452492 and EAGER Award #1644363), and the NCI (CA207407). M. Stephan was also supported by a Research Scholar Grant (RSG-16-110-01– LIB) from the American Cancer Society, a 2018 Emerging Leader Award from the Mark Foundation for Cancer Research, and a 2018 Investigator Award from the Alliance for Cancer Gene Therapy (ACGT). This research was also supported by an Allen Distinguished Investigator Award, a Paul G. Allen Frontiers Group advised grant of the Paul G. Allen Family Foundation.

## Author contributions

N.N.P. helped conceive the study and designed and performed the experiments. S.B.S. produced polymers and antibody-PGA conjugates. A.L.K. performed and analyzed the in vivo safety/toxicity studies. P.S.N. provided antigen expression profiles for human prostate cancer metastases showing the diversity of antigen expression. M.T.S. conceived the study, helped design the experiments, and wrote the manuscript.

## Competing interests

M.T.S. is a consultant of Tidal Therapeutics and holds stocks in the company. The remaining authors declare no competing interests.
