## [Peer Review File · Nature Communications]

Reviewers' comments:

Reviewer #1, expert in RNA delivery in vivo (Remarks to the Author):

This is an interesting manuscript demonstrating impressive and innovative data. Yet, in my opinion, this study can be complemented with a few more experiments some crucial.

1. Particle biodistribution:

The authors should do a whole-body biodistribution of these NPs (Liver, Lung, Spleen, kidneys, etc.). Perhaps by implementing luciferase-based or a Cre mRNA-tdtomato bio distribution assay.

2. Efficacy in a syngeneic immune-competent mouse model

Since this is an immunotherapy approach the authors must demonstrate the efficacy of the mRNA based CART treatment in an immunocompetent mouse model.

a. This will determine whether this strategy is effective in the presence of the immune-suppressive TME, a known challenge in progressing CART treatments for solid cancers. Overall this will better reflect the human clinical situation compared to the xenograft models described.

b. This will also determine if lymphodepletion is required prior to mRNA-NP CAR therapy. Studies demonstrate that lymphodepleting preconditioning improves CAR T cell therapy. Furthermore, lymphodepletion prior to treatment was reported to mitigate immune related toxicities in patients.

Currently, patients undergo leukapheresis prior to lymphodepleting chemotherapy preconditioning. The authors state that it is "uncomfortable for the patient, incurs a substantial monetary cost, and ultimately may be rate-limiting for large-scale adoption of autologous T cell".

Therefore, to avoid leukapheresis, the mRNA-NP CAR platform must demonstrate efficacy without lymphodepletion in an immunocompetent model.

Otherwise the lymphodepleting chemotherapy regimen will not leave any T cells to target in-vivo and will be problematic in a clinical setting employing these NPs since the T cells need to be present in order to deliver the CAR mRNA to them.

3. NP stability in serum

The authors should show more stability of these NPs in serum. Perhaps by incubating NPs in serum at 37°C and testing encapsulation.

4. anti-CD8 coupling. How many mAbs are coated on the LNPs ? how reproducible is this strategy ? it seems that it will be very hard to scale up such a strategy.

5. Nanoparticle transfection of ex vivo- cultured T cells

The authors transfect ex-vivo cultured isolated CD8+ T cells. It would be interesting to

demonstrate that these particles do in fact selectively bind and transfect only CD8+ T cells in-vitro, perhaps by incubating with both CD8+ and CD4+ cells. or in a mix population experiment.

6. Methods section

Under – "Retroviral vectors and virus production"

" . It consists of a single-chain antibody targeting mouse ROR1 or EGFRvIII linked via a c-myc tag to a synthetic receptor skeleton comprised of the CD8 hinge, the CD28 transmembrane and signaling domains, and the signaling domain from CD3 ζ ." Does the CAR target mouse ROR1? Is it cross-reactive with human ROR1? Or a typo?

Reviewer #2, expert in CAR-T cells (Remarks to the Author):

This manuscript is overall well written. Figure 1 and 2 are beautifully done and are like artworks. The authors have great initiative to program T cells in vivo. If successful, the in vivo transducing T cell methods will significantly reduce the cost for T cells engineering.

However, this work is mainly descriptive and lacks mechanistic insight. For example, there is no study on how the polymeric nanocarriers were breaking down, and how the mRNA inside the nanoparticles were incorporated into the T cells and why the CAR/TCR expression of the surface expression was transient. In fact, throughout the whole paper, there is barely any mechanism investigated.

The work was carried out in the NSG model system, that does not capture some important human settings. In this experimental system, only human T cell were adoptively transferred. The nanoparticles contained anti-CD8 antibodies on their surface. Therefore, the nanoparticles could attach to the CD8 human T cells easily in this artificial system. In patient settings, there are many cell populations, including other cells expressing CD8, and phagocytes. It is therefore a lot more difficult for this method to work. This paper therefore, failed to demonstrate the translational value.

The toxicity study of the nanoparticle should include T cells. Even the nanoparticles by themselves are safe, the biggest concern in T cell therapy field is in fact the toxicity caused by the T cells. In addition, as mentioned above, in this NSG system, only T cells were adoptively transferred, and other cell component such as macrophages, NK cells and dendritic cells were lacking, ADCC and other toxicity related issues could not be addressed in this model system.

A number of the experiments carried out lacked appropriate controls. For example, Figure 3d, 3e, 3i and 3j should use non-transduced T cells as experimental controls.

Figure 4 should add human T cells (no transduction) group, and CAR+ control NP group as the experiment controls. Same to Figure 5.

The whole paper lacks a control RNA carried by nanoparticles as an experimental control.

Reviewer #3, expert in CAR-T cells (Remarks to the Author):

N.N. Parayath et al explored the use of IVT mRNA nanocarriers as an injectable drug to genetically reprogram circulating CD8+ T cells to transiently express tumor antigen-specific receptors. The authors have introduced both CAR and TCR mRNA constructs into the T cells. As mentioned in the introduction, the presented nanocarrier technology may facilitate the production of redirected T cells for adoptive immunotherapy. The authors describe that CD8+ T cells can be specifically equipped with CAR and TCR constructs in vitro as well as in vivo. However, some major points in the manuscript should be clarified in order to correctly characterize the applicability of the presented technology.

1. mRNA transfer using nanocarriers (in vitro experiments).

The authors describe that the nanocarriers are associated with ligands to specifically bind cytotoxic T cells. In this case, a CD8-specific antibody is used to recognize and bind CD8+ T cells and introduce the mRNA load. Correspondingly, in Figures 3A,3F the authors demonstrate the presence of the transferred 1928 CAR mRNA in human CD3+ T cells and in Figure 3C,3H the corresponding expression on the T cell surface. However, there is no investigation as to whether the nanocarriers can fuse with the cell membrane without ligand binding. This would require a control experiment in which nanocarriers will be loaded with mRNA and a ligand (antibody) of irrelevant specificity. This would enable us to find out whether transfer into cells is possible even without ligand-receptor binding.

Furthermore, the authors should present the expression level of CAR and TCR constructs in CD8+ as well as CD4+ T cells if CD8-specific nanocarriers were used. This would also demonstrate the specificity of the approach in vitro.

The authors present in Figures 3E and 3J using IL-2, TNF- α and IFN- γ secretion that the nanocarrier technology provides equivalent effector cells compared to retrovirally transduced T cells. It would be very useful if the authors would extend the experiments with control groups: mock-transduced T cells and T cells with irrelevant specificity (Figure 3D, 3E, 3F and 3J).

It would be very useful if the authors would give the exact effector and target cell numbers in the cytotoxic studies (methods).

2. mRNA transfer using nanocarriers (in vivo experiments).

The authors apply nanocarrier technology in immunodeficient mouse models (NSG) and compare the efficiency of reprogrammed T cells with T cells redirected using retroviral gene transfer.

1. CD19+ Leukemia mouse model

The authors present that CD19-specific CAR T cells, regardless of the type of gene transfer technology, increase the survival of the tumor bearing mice. Using three examples representing a larger group of animals (Fig 4D), the authors show the CAR expression on CD8+ T cells at different time points of the animal experiment before and after infusion of nanocarriers loaded with CD19 CAR mRNA. In principle, the specificity of the nanocarrier technology should also be asked here. Both CD4 and CD8 T cells should be examined for CAR expression. It would definitely not be an advantage if the CARs or TCRs were expressed in cell types other than the desired ones.

A simple mouse model would be best, where the authors insert GFP mRNA into CD8-specific nanocarriers and inject the carriers into immunocompetent mice. Subsequently, immune cells (T-, B-, NK, DC, . . .) and certain tissue cells (lung, liver, brain, kidney, . . .) must be tested for GFP expression. Only when the specificity of the approach has been clarified does it make sense to think about applying this technology elsewhere.

2. ROR1+ Prostate tumors

As in the previous mouse experiment: Please first prove the specificity of your gene transfer method (CD4+, CD8+,...).

3. HBV-specific TCR T cells

Unfortunately, this approach also lacks proof of specificity for presented gene transfer method.

The toxicity studies were shown in Figures 7 and 8. But, the initial aim is to investigate how T cells react to multiple administration of nanocarriers. Does repeated administration of the nanocarrier

lead to apoptosis of the T cell ? T cell phenotype changing? T cell proliferation ? The authors should first answer the more obvious questions for specificity of presented technology and for T cell biology.

At this time point, I have great concerns that the presented work will meet the qualitative standards of Nature Communications.

Reviewer #1 (Remarks to the Author):

1. Particle biodistribution:

The authors should do a whole-body biodistribution of these NPs (Liver, Lung, Spleen, kidneys, etc.). Perhaps by implementing luciferase-based or a Cre mRNA-tdtomato bio distribution assay.

We now include a biodistribution study using Gt(ROSA)26Sortm14(CAG-tdTomato)Hze/J (Ai reporter) mice as suggested by the reviewer. We first measured whole-organ fluorescence in Ai14 mice following a single injection of lymphocyte-targeted (or isotype control-functionalized) nanoparticles carrying Cre mRNA. The highest gene expression mediated by non-targeted particles was found in the liver, while lymphocyte-targeted nanocarriers induced gene transfer mainly in the spleen, lymph nodes, and thymus (Fig. 1 below, new **Fig. 5a, b** in our revised manuscript). A detailed flow cytometry analysis of the spleen (new **Fig. 5c**) revealed that CD3-targeted nanoparticles preferentially transfected T cells ($8.1\% \pm 1.9\%$), without compromising viability (new **Supplementary Fig. 2**). Much lower levels of dtTomato signals were detected in other CD45+ (immune) subtypes, such as macrophages ($3.2\% \pm 1.5\%$), B cells ($1.1\% \pm 0.9\%$), neutrophils ($0.3\% \pm 0.2\%$), and dendritic cells ($1.9\% \pm 0.8\%$).

Figure 1: Efficient T-cell targeting in immunocompetent mice. B6.Cg-Gt(ROSA)26Sortm14(CAG-tdTomato)Hze/J (Ai reporter) mice were injected intravenously with a single dose of nanoparticles loaded with 15 μ g mRNA encoding nuclear localization signal (NLS)-Cre. Nanoparticles were targeted to mouse T cells using a full-length anti-CD3 MulgG2a, or IgG2a isotype control. Both antibodies were designed as LALAPG variants to ablate Fc receptor binding and complement activation. Forty-eight hours after injection, organs were collected and whole-organ dtTomato fluorescence was measured using fluorescent IVIS imaging. Single-cell suspensions of spleens and blood were labeled with antibodies against various immune cell subtypes and analyzed by flow cytometry.

(a) Representative (n=7, 1 pictured) dtTomato expression in organs under fluorescent IVIS imaging. (b) Quantification of fluorescent signal in each organ. Each symbol indicates one measured organ. Horizontal lines indicate mean values, and error bars represent standard deviation of the mean. Pairwise differences in fluorescent intensities between the groups were statistically analyzed with the Wilcoxon rank-sum test (two-sided). (c) Graph displaying the mean \pm SD percent of immune CD45+ dtTomato+ cell types in the spleen. Macrophages (CD45+, CD11b+, MHCII+, CD11c-, Ly6C-/low, Ly6G-), B cells (CD45+, B220+), T cells [CD4+ T cells (CD45+, TCR- β chain+, CD4+, CD8-), CD8+ T cells (CD45+, TCR- β chain+, CD4-, CD8+)], neutrophils (CD45+, CD11b+, MHCII+, CD11c-, Ly6G+), and dendritic cells (CD45+, CD11c+, CD11b-, MHCII+) were measured. Each symbol indicates one mouse. Horizontal lines indicate mean values, and error bars represent standard deviation of the mean. Pairwise differences in fluorescent intensities between the groups were statistically analyzed with the Wilcoxon rank-sum test (two-sided). N=7 biologically independent samples.

2. Efficacy in a syngeneic immune-competent mouse model

Since this is an immunotherapy approach the authors must demonstrate the efficacy of the mRNA based CART treatment in an immunocompetent mouse model. This will determine whether this strategy is effective in the presence of the immune-suppressive TME, a known challenge in progressing CART treatments for solid cancers. Overall this will better reflect the human clinical situation compared to the xenograft models described.

Following the reviewer's comment, we systemically injected luciferase-expressing E μ -ALL01 leukemia cells (an immunocompetent mouse model of B cell acute lymphoblastic leukemia that recapitulates the disease at genetic, cellular, and pathologic levels¹) into albino C57BL/6 mice and used bioluminescent imaging to quantify differences in tumor progression between treatment groups (Figure 2 below, updated **Fig. 6a** of our revision). Mice received either CD3-targeted nanoparticles delivering mRNA encoding the fully murine 1928z CAR or GFP control. A third group received no treatment. We found that only injections of nanoparticles encoding the 1928z CAR effectively controlled leukemia progression (new **Fig. 6b, c**), resulting in an average 26-fold reduced tumor burden after three weeks of therapy compared to GFP controls.

- This will also determine if lymphodepletion is required prior to mRNA-NP CAR therapy. Studies demonstrate that lymphodepleting preconditioning improves CAR T cell therapy. Furthermore, lymphodepletion prior to treatment was reported to mitigate immune related toxicities in patients.

The new data generated in fully immunocompetent mice described above (Questions 1 and 2) establish that mRNA-NP CAR therapy can *in situ* program sufficient numbers of host T cells (average 8.1%) with disease-specific CARs to stop tumor progression (**Fig. 1c**, **Fig. 2**). Since this treatment strategy relies on the presence of circulating T cells, a preconditioning chemotherapy to deplete endogenous lymphocytes becomes obsolete. Chemotherapy pretreatment is a costly and burdensome hospital procedure, and has too many risks for use in non-oncology indications. Thus, we anticipate that our approach can transform T-cell therapy into a patient-friendly treatment for a wide range of diseases, available at the day of diagnosis and as frequently as medically necessary.

- NP stability in serum. The authors should show more stability of these NPs in serum. Perhaps by incubating NPs in serum at 37°C and testing encapsulation.

We repeated our mRNA encapsulation experiments and confirmed that PBAE-PGA-CD3 nanoparticles very efficiently encapsulate mRNA (close to 98%, **Fig. 3** below). Encapsulation,

was measured with a Qubit RNA HS assay kit (mRNA encapsulated = (total mRNA in sample – free mRNA detected by Qubit)/total mRNA in sample). It should be pointed out that we cannot isolate/spin down the nanoparticles, so our calculations were based on the free mRNA detected in the supernatant. We tried measuring free mRNA over time in serum; however, the unstable nature of mRNA and its susceptibility to degradation by RNases made it impossible to get a reliable readout. We know from the literature that PBAE 447 is highly biodegradable, with a serum half-life between 1 and 7 hours in aqueous conditions². This time frame is ideal for mRNA gene therapy into circulating T cells, as the polymer condenses and effectively protects nucleic acids against degradation while they are within the endosome, but releases them once the nanoparticles come in contact with the cytoplasm.

5. Anti-CD8 coupling. How many mAbs are coated on the LNPs ? how reproducible is this strategy ? it seems that it will be very hard to scale up such a strategy.

To develop a robust approach to scale-up production of genetic lymphocyte-programming nanoparticles to support Phase 1 clinical trials, Fred Hutchinson Cancer Research Center spun out Tidal Therapeutics (<https://labcentral.org/resident-companies/tidal-therapeutics>). Below is a summary of the confidential information and calculations Tidal Therapeutics provided to us to address the Reviewer's question regarding scale-up manufacturing for clinical testing:

SCALE-UP

NP Protocol for Phase 1/2 Manufacturing.

The micro-scale process used at the research stage has been translated (Stephan Lab, Fred Hutch) to a scale suitable for manufacture of drug for Phase 1/2 clinical trials. Process 2 uses continuous flow micro-mixing using micro-fluidic chipset(s). Reagent concentrations and ratio of PBAE to mRNA have been maintained at levels used in Process 1.

Each chipset consists of two micro-fluidic chips and three syringe pumps to provide precise reagent flows. Each chipset (mixing chips and associated tubing) is provided as pre-sterilized single-use assemblies, thus eliminating the need for cleaning and sterilization requirements in cGMP manufacturing. Depending on drug product requirements, throughput can be adjusted using multiple parallel chipsets. A typical single chipset configuration is shown in Figure 4 below.

Process 2, Step 2: Concentration and Diafiltration using Tangential Flow Ultra-Filtration.

After formation of the nanoparticles, residual reagents are removed, and the nanoparticles are concentrated and then diafiltered into formulation buffer using ultrafiltration. Following concentration/diafiltration, the bulk drug is sterile-filtered and kept frozen at -20C till the final fill-finish step.

Phase 1 T-cell targeted trial Drug Requirements, Analysis of Process 2 throughput & Costs for Phase 1 Clinical Manufacturing.

Preliminary plans for a Phase 1 trial call for approximately 12 patients administered up to a maximum total dose of 5 mg/week with 9 weeks of dosing, working out to a total amount of NPs containing 540 mg of mRNA. Adding in 50% overage (to account for process losses and drug substance testing), we arrive at a drug substance requirement of approximately 800 mg. Using the current reagent concentrations, we can calculate process throughputs under different process configurations. Table 1 shows the results of these calculations. Using 8 high-flow chipsets, Step 1 of the process can be completed in 1.67 hours. If necessary, 2 batches of NPs can be produced in 4 hours. Filters can be sized to complete Step 2 of the process in 2 hours. In summary, drug substance required for 24 patients at a max dose of 5 mg/week — 9 weeks' total dosing — can be manufactured in a single 8-hour manufacturing shift.

	Concentration (mg/ml)	Volume (ml)	Low-Flow Chipset Flow Rate (ml/min)	High-Flow Chipset Flow Rate (ml/min)
mRNA	0.1	8000	3	10
PBAE	6	8000	3	10
PGA-CD3	0.25	16000	6	20
# of Chip sets	1	4	8	
Process Time @ Low Flow (hours)	44	11	6	
Process Time @ High Flow (hours)	13.33	3.33	1.67	

6. Nanoparticle transfection of ex vivo- cultured T cells

The authors transfect ex-vivo cultured isolated CD8+ T cells. It would be interesting to demonstrate that these particles do in fact selectively bind and transfect only CD8+ T cells in-vitro, perhaps by incubating with both CD8+ and CD4+ cells. or in a mix population experiment.

We know from our previous studies, where we used chemically identical nanocarriers to deliver genome editing mRNA into cultured T cells or hematopoietic stem cells³, that our antibody-targeted nanoparticles selectively bind target cells, with low off-target cell transfection (see Fig 2, Suppl. Fig. 3 and Fig. 6 below).

Article | Open Access | Published: 30 August 2017

Hit-and-run programming of therapeutic cytoreagents using mRNA nanocarriers

H. F. Moffett, M. E. Coon, S. Radtke, S. B. Stephan, L. McKnight, A. Lambert, B. L. Stoddard, H. P. Kiem & M. T. Stephan 
Nature Communications **8**, Article number: 389 (2017) | Cite this article

2253 Accesses | 24 Citations | 104 Altmetric | Metrics

NATURE COMMUNICATIONS | DOI: 10.1038/s41467-017-00505-8 ARTICLE

Fig. 2 mRNA nanoparticle transfection choreographs robust transgene expression by lymphocytes. **a** Primary T-cells were mixed with CD3-targeted polymeric nanoparticles (NPs) carrying Cy5-labeled mRNA. Confocal microscopy establishes that these particles are rapidly internalized from the cell surface. The images are representative of 15 randomly chosen fields. Scale bars, 2 μ m. **b** Flow cytometry of preactivated PBMCs 24 h after incubation with CD3-targeted or isotype control antibody-targeted nanoparticles bearing eGFP-encoding mRNA. **c** Bar graph summarizing transfection efficiencies from three independent experiments conducted in duplicate.

Supplementary Fig. 3 CD3-mediated targeting confines nanoparticle interactions to T cells. Unstimulated bulk peripheral blood mononuclear cells (PBMC) were directly incubated with CD3-targeted NPs carrying GFP mRNA. One day later, we compared transfection efficiencies in T cells versus B cells. Flow cytometry plots are shown in (a). (b) Summary plot showing mean transfection efficiency and S.E.M. of three independent experiments conducted in duplicate.

Fig. 6

From: Hit-and-run programming of therapeutic cytoagents using mRNA nanocarriers

Fig. 6 a Targeting of CD105 enables specific transfection of HSC CD34⁺ cells. Cells were left untreated, or transfected with eGFP-encoding mRNA in nanoparticles (NPs) coated with PGA coupled to a control antibody or anti-CD105. Transfection efficiency was assayed by flow cytometry after 24 h. b NP transfection efficiency in CD34⁺ samples from three independent donors. Viability is shown in c. d

To confirm these studies, we are now including in our revised manuscript *in vitro* and *in vivo* experiments that specifically measure T-cell targeting. For these experiments, we functionalized nanoparticles with either anti-CD3 antibodies or an isotype control.

For *in vitro studies*, which are now summarized as **Supplementary Fig. 1** (Fig A below), we transfected human T cells with various concentrations of mRNA nanoparticles encoding Firefly luciferase. Gene expression was measured 24 hours after transfection using IVIS imaging. We found that only nanoparticles functionalized with anti-CD3 antibodies efficiently delivered the transgene, whereas isotype control-functionalized nanoparticles yielded gene expression close to background levels.

7. Methods section

Under – "Retroviral vectors and virus production " It consists of a single-chain antibody targeting mouse ROR1 or EGFRvIII linked via a c-myc tag to a synthetic receptor skeleton comprised of the CD8 hinge, the CD28 transmembrane and signaling domains, and the signaling domain from CD3 ζ ." Does the CAR target mouse ROR1? Is it cross-reactive with human ROR1? Or a typo?

We corrected this typo. This CAR targets human ROR1.

We appreciate the constructive criticisms, and hope that our responses appropriately address the issues the reviewer raised. The changes made in response to them have substantially improved our manuscript.

References

1. Davila, M.L., Kloss, C.C., Gunset, G. & Sadelain, M. CD19 CAR-targeted T cells induce long-term remission and B Cell Aplasia in an immunocompetent mouse model of B cell acute lymphoblastic leukemia. *PLoS One* **8**, e61338 (2013).
2. Mangraviti, A., *et al.* Polymeric nanoparticles for nonviral gene therapy extend brain tumor survival in vivo. *ACS Nano* **9**, 1236-1249 (2015).
3. Moffett, H.F., *et al.* Hit-and-run programming of therapeutic cytoreagents using mRNA nanocarriers. *Nat Commun* **8** (2017).

Reviewer #2 (Remarks to the Author):

This manuscript is overall well written. Figure 1 and 2 are beautifully done and are like artworks. The authors have great initiative to program T cells in vivo. If successful, the in vivo transducing T cell methods will significantly reduce the cost for T cells engineering.

1. However, this work is mainly descriptive and lacks mechanistic insight. For example, there is no study on how the polymeric nanocarriers were broken down, and how the mRNA inside the nanoparticles were incorporated into the T cells and why the CAR/TCR expression of the surface expression was transient. In fact, throughout the whole paper, there is barely any mechanism investigated.

How are the polymeric nanocarriers broken down? The “PBAE-447” polymer we are using in our study to condense mRNA into nanoparticles was originally developed in 2011 by the Jordan Green lab at Johns Hopkins University¹. Over the past decade, his group and others have extensively characterized the key properties of PBAE (reviewed in 2020 by Karlsson et al.²).

- PBAEs enable endosomal escape by undergoing protonation at the lower pH of the endosomal compartment, leading to osmotic pressure buildup due buffering, which causes endosomal disruption. High-throughput combinatorial library screens of PBAEs for nucleic acid delivery have shown that the presence of tertiary amines improves buffering capacity at low pH and facilitate endosomal escape.
- The ester bonds in the backbone structure of PBAEs undergo hydrolysis in aqueous conditions, making PBAEs less toxic than other non-degradable cationic polymers, such as PEI, which has been broadly investigated as a nucleic acid delivery vehicle.

How is the mRNA inside the nanoparticles incorporated into the T cells? We know from our previous studies, where we used chemically identical nanocarriers to deliver genome editing mRNA into cultured T cells or hematopoietic stem cells³, that our antibody-targeted nanoparticles selectively bind target cells, and initiates rapid receptor-induced endocytosis to get internalized (see Fig. 1 below).

Hit-and-run programming of therapeutic cytoreagents using mRNA nanocarriers

H. F. Moffett, M. E. Coon, S. Radtke, S. B. Stephan, L. McKnight, A. Lambert, B. L. Stoddard, H. P. Kiem & M. T. Stephan 
Nature Communications **8**, Article number: 389 (2017) | Cite this article

2353 Accesses | **24** Citations | **104** Altmetric | Metrics

Fig. 1 mRNA nanoparticle transfection choreographs robust transgene expression by lymphocytes. **a** Primary T-cells were mixed with CD3-targeted polymeric nanoparticles (NPs) carrying Cy5-labeled mRNA. Confocal microscopy establishes that these particles are rapidly internalized from the cell surface. The images are representative of 15 randomly chosen fields. *Scale bars*, 2 μ m. **b** Flow cytometry of preactivated PBMCs 24 h after incubation with CD3-targeted or isotype control antibody-targeted nanoparticles bearing eGFP-encoding mRNA. **c** *Bar graph* summarizing transfection efficiencies from three independent experiments conducted in duplicate.

Why the CAR/TCR expression of the surface expression was transient? The nanoparticles we used in our study deliver mRNA encoding CAR transgenes. mRNA is naturally transiently active in the cytosol and does not integrate into the host genome.

2. The work was carried out in the NSG model system, that does not capture some important human settings. In this experimental system, only human T cell were adoptively transferred. The nanoparticles contained anti-CD8 antibodies on their surface. Therefore, the nanoparticles could attach to the CD8 human T cells easily in this artificial system. In patient settings, there are many cell populations, including other cells expressing CD8, and phagocytes. It is therefore a lot more difficult for this method to work. This paper therefore, failed to demonstrate the translational value.

Following the reviewer's comment, we systemically injected luciferase-expressing E μ -ALL01 leukemia cells (an immunocompetent mouse model of B cell acute lymphoblastic leukemia that recapitulates the disease at genetic, cellular, and pathologic levels⁴) into albino C57BL/6 mice and used bioluminescent imaging to quantify differences in tumor progression between treatment groups (Figure 2 below, updated **Fig. 6a** of our revision). Mice either received CD3-targeted nanoparticles delivering mRNA encoding the fully murine 1928z CAR, or GFP control. A third group received no treatment. We found that only injections of nanoparticles encoding the 1928z CAR effectively controlled leukemia progression (new **Fig. 6b, c**), resulting in an average 26-fold reduced tumor burden after three weeks of therapy compared to GFP controls.

3. The toxicity study of the nanoparticle should include T cells. Even the nanoparticles by themselves are safe, the biggest concern in T cell therapy field is in fact the toxicity caused by the T cells. In addition, as mentioned above, in this NSG system, only T cells were adoptively transferred, and other cell component such as macrophages, NK cells and dendritic cells were lacking, ADCC and other toxicity related issues could not be addressed in this model system.

Conventional T-cell therapy lymphodepletes patients, which opens up cytokine niches in primary and secondary lymphoid organs of the host. As a result, the fully activated ex vivo engineered CAR T-cells swiftly engraft and rapidly expand. As shown in the figure on the right, this

expansion can be more than 3-log over one week, when infusing CD-19 targeted CAR-T cells into patients with leukemia. As the Reviewer pointed out, some patients develop the cytokine release syndrome (CRS) and CAR-T cell-related neurotoxicity, and require treatment with corticosteroids and cytokine-blocking antibodies. It should be pointed out that T-cell mediated toxicities have not been reported when (1) patients were not lymphodepleted⁵, or (2) patients with solid tumors were treated with

CAR T cells⁶ – even when directly injecting CAR T cells into the CNS⁷. Using our technology, the host must not be lymphodepleted, as the goal is to reprogram circulating T cells. Cytokine storms are therefore unlikely. Ultimately, toxicology studies in large animals (rhesus macaques) are required to confirm safety of this T-cell reprogramming nanodrug, since rodents cannot faithfully recapitulate T-cell associated CRS and neurotoxicity.⁸

Antibody-dependent cell-mediated cytotoxicity (ADCC), as suggested by the Reviewer, can also be ruled out as source of toxicity in our platform. The T-cell targeting ligands (anti-CD3 or anti-CD8) we used on the surface of nanoparticles were deglycosylated in the C γ 2 domain of the Fc region. Deglycosylated IgG no longer bind significantly to Fc γ Rs and to C1q, thus being unable to trigger ADCC and complement activation⁹.

4. A number of the experiments carried out lacked appropriate controls. For example, Figure 3d, 3e, 3i and 3j should use non-transduced T cells as experimental controls.

To demonstrate specificity for the CD19 and HBcore18-27 antigens, we now include control groups in all tumor killing assays and cytokine release measurements shown in the revised **Fig. 3** (Fig. 3 below). In CAR experiments, we transduced T cells with control lentivirus, encoding the

P28z CAR, which is targeting the Prostate-Specific Membrane Antigen¹⁰. In TCR engineering experiments, we included T cells that we transduced with TCRs specific for human Mesothelin¹¹. We also included more detailed information regarding T cell/tumor cell ratios to the figure legend.

Fig 3 (a) *In vitro* assay comparing cytotoxicity of nanoparticle- vs. retrovirus-transfected T cells against Raji lymphoma cells. T cells were co-cultured with Raji tumor cells at a 5:1 ratio. We used the IncuCyte Live Cell Analysis System to quantify immune cell killing of Raji Nuclight Red cells by 1928z-CAR or control (P28z-CAR) -transfected T cells over time. Data are representative of two independent experiments. Each point represents the mean \pm s.e.m. pooled from two independent experiments conducted in triplicate. (b) ELISA measurements of IL-2 (at 24 h) and TNF- α and IFN- γ (at 48 h) secretion by transfected cells. (c) Cell killing of HepG2-core Nuclight Red cells by HBcore18-27 or control (MSLN-) TCR-transfected T cells over time. T cells were co-cultured with HepG2 tumor cells at a 5:1 ratio. (d) ELISA measurements of cytokine secretion by transfected cells.

5. Figure 4 should add human T cells (no transduction) group, and CAR+ control NP group as the experiment controls.

We now included these two treatment groups in our revised manuscript (Figure 4 below, updated **Fig. 4** of our revision).

Figure 4: Nanoparticle-programmed CAR lymphocytes can cause leukemia regression with efficacies similar to adoptive T-cell therapy (a) Time line and nanoparticle (NP) dosing regimen. (b) Sequential bioluminescence imaging of firefly luciferase-expressing Raji lymphoma cells systemically injected into NSG mice. Five representative mice from each cohort ($n = 10$) are shown. (c) Survival of animals following therapy, depicted as Kaplan-Meier curves. Shown are ten mice per treatment group pooled from three independent experiments. ms, median survival. Statistical analysis between the treated experimental and the untreated control group was performed using the Log-rank test; $P < 0.05$ was considered significant. (d) Flow cytometry of peripheral T cells before and after injection of nanoparticles delivering IVT mRNA that encodes the 1928z CAR. The three profiles for each time point shown here are representative of two independent experiments consisting of ten mice per group. (e) Overview graph displaying the percentages of CAR-transfected CD8+ T cells following repeated infusion of 1928z CAR NPs. Every line represents one animal. Shown are ten animals pooled from two independent experiments. Mean transfection efficiencies (\pm SD) for each time point are shown at the top.

6. The whole paper lacks a control RNA carried by nanoparticles as an experimental control.

We did use relevant GFP control mRNA treatment arms in our originally submitted studies (summarized in the figure below). PBAE nanoparticles do not form without adding mRNA, so “control NPs” refers to PBAE nanoparticles loaded with GFP mRNA).

We appreciate the constructive criticisms from this reviewer, and hope that our responses satisfactorily address the issues they raised.

References

1. Tzeng, S.Y., *et al.* Non-viral gene delivery nanoparticles based on poly(beta-amino esters) for treatment of glioblastoma. *Biomaterials* **32**, 5402-5410 (2011).
2. Karlsson, J., Rhodes, K.R., Green, J.J. & Tzeng, S.Y. Poly(beta-amino ester)s as gene delivery vehicles: challenges and opportunities. *Expert Opin Drug Deliv* (2020).
3. Moffett, H.F., *et al.* Hit-and-run programming of therapeutic cytoreagents using mRNA nanocarriers. *Nat Commun* **8**(2017).
4. Davila, M.L., Kloss, C.C., Gunset, G. & Sadelain, M. CD19 CAR-targeted T cells induce long-term remission and B Cell Aplasia in an immunocompetent mouse model of B cell acute lymphoblastic leukemia. *PLoS One* **8**, e61338 (2013).
5. Kochenderfer, J.N., *et al.* Eradication of B-lineage cells and regression of lymphoma in a patient treated with autologous T cells genetically engineered to recognize CD19. *Blood* **116**, 4099-4102 (2010).
6. Haas, A.R., *et al.* Phase I Study of Lentiviral-Transduced Chimeric Antigen Receptor-Modified T Cells Recognizing Mesothelin in Advanced Solid Cancers. *Mol Ther* **27**, 1919-1929 (2019).
7. Bagley, S.J., Desai, A.S., Linette, G.P., June, C.H. & O'Rourke, D.M. CAR T-cell therapy for glioblastoma: recent clinical advances and future challenges. *Neuro Oncol* **20**, 1429-1438 (2018).
8. Taraseviciute, A., *et al.* Chimeric Antigen Receptor T Cell-Mediated Neurotoxicity in Nonhuman Primates. *Cancer Discov* **8**, 750-763 (2018).
9. Abes, R. & Teillaud, J.L. Impact of Glycosylation on Effector Functions of Therapeutic IgG. *Pharmaceuticals (Basel)* **3**, 146-157 (2010).
10. Gade, T.P., *et al.* Targeted elimination of prostate cancer by genetically directed human T lymphocytes. *Cancer Res* **65**, 9080-9088 (2005).
11. Stromnes, I.M., *et al.* T Cells Engineered against a Native Antigen Can Surmount Immunologic and Physical Barriers to Treat Pancreatic Ductal Adenocarcinoma. *Cancer Cell* **28**, 638-652 (2015).

Reviewer #3 (Remarks to the Author):

1. mRNA transfer using nanocarriers (in vitro experiments).
The authors describe that the nanocarriers are associated with ligands to specifically bind cytotoxic T cells. In this case, a CD8-specific antibody is used to recognize and bind CD8+ T cells and introduce the mRNA load. Correspondingly, in Figures 3A,3F the authors demonstrate the presence of the transferred 1928 CAR mRNA in human CD3+ T cells and in Figure 3C,3H the corresponding expression on the T cell surface. However, there is no investigation as to whether the nanocarriers can fuse with the cell membrane without ligand binding. This would require a control experiment in which nanocarriers will be loaded with mRNA and a ligand (antibody) of irrelevant specificity. This would enable us to find out whether transfer into cells is possible even without ligand-receptor binding.

We know from our previous studies, where we used chemically identical nanocarriers to deliver genome editing mRNA into cultured T cells or hematopoietic stem cells¹, that our antibody-targeted nanoparticles selectively bind target cells, with low off-target cell transfection (see Fig 2, Suppl. Fig. 3 and Fig. 6 below).

 Article | Open Access | Published: 30 August 2017

Hit-and-run programming of therapeutic cytoreagents using mRNA nanocarriers

H. F. Moffett, M. E. Coon, S. Radtke, S. B. Stephan, L. McKnight, A. Lambert, B. L. Stoddard, H. P. Kiem & M. T. Stephan 
Nature Communications **8**, Article number: 389 (2017) | Cite this article

2353 Accesses | 24 Citations | 104 Altmetric | Metrics

NATURE COMMUNICATIONS | DOI: 10.1038/s41467-017-00505-8 ARTICLE

Fig. 2 mRNA nanoparticle transfection choreographs robust transgene expression by lymphocytes. **a** Primary T-cells were mixed with CD3-targeted polymeric nanoparticles (NPs) carrying Cy5-labeled mRNA. Confocal microscopy establishes that these particles are rapidly internalized from the cell surface. The images are representative of 15 randomly chosen fields. *Scale bars*, 2 μm . **b** Flow cytometry of preactivated PBMCs 24 h after incubation with CD3-targeted or isotype control antibody-targeted nanoparticles bearing eGFP-encoding mRNA. **c** *Bar graph* summarizing transfection efficiencies from three independent experiments conducted in duplicate.

Supplementary Fig. 3 CD3-mediated targeting confines nanoparticle interactions to T cells. Unstimulated bulk peripheral blood mononuclear cells (PBMC) were directly incubated with CD3-targeted NPs carrying GFP mRNA. One day later, we compared transfection efficiencies in T cells versus B cells. Flow cytometry plots are shown in (a). (b) Summary plot showing mean transfection efficiency and S.E.M. of three independent experiments conducted in duplicate.

Fig. 6

From: Hit-and-run programming of therapeutic cytoagents using mRNA nanocarriers

Fig. 6 **a** Targeting of CD105 enables specific transfection of HSC CD34⁺ cells. Cells were left untreated, or transfected with eGFP-encoding mRNA in nanoparticles (NPs) coated with PGA coupled to a control antibody or anti-CD105. Transfection efficiency was assayed by flow cytometry after 24 h. **b** NP transfection efficiency in CD34⁺ samples from three independent donors. Viability is shown in **c**. **d**

To confirm these studies, we are now including in our revised manuscript *in vitro* and *in vivo* experiments that specifically measure T-cell targeting. For these experiments, we either functionalized nanoparticles with anti-CD3 antibodies or isotype control.

For *in vitro studies*, which are now summarized as **Supplementary Fig. 1** (Fig A below), we transfected human T cells with various concentrations of mRNA nanoparticles encoding Firefly luciferase. Gene expression was measured 24 hours after transfection using IVIS imaging. We found that only nanoparticles functionalized with anti-CD3 antibodies efficiently delivered the transgene, whereas isotype control-functionalized nanoparticles yielded gene expression close to background levels.

In vivo studies are described below as part of our reply to Question 3.

2. The authors present in Figures 3E and 3J using IL-2, TNF- α and IFN- γ secretion that the nanocarrier technology provides equivalent effector cells compared to retrovirally transduced T cells. It would be very useful if the authors would extend the experiments with control groups: mock-transduced T cells and T cells with irrelevant specificity (Figure 3D, 3E, 3F and 3J). It would be very useful if the authors would give the exact effector and target cell numbers in the cytotoxic studies (methods).

To demonstrate specificity for the CD19 and HBcore18-27 antigens, we now include control groups in all tumor killing assays and cytokine release measurements shown in the revised Fig. 3 (Fig. B below). In CAR experiments, we transduced T cells with control lentivirus, encoding the P28z CAR, which is targeting the Prostate-Specific Membrane Antigen². In TCR engineering experiments, we included T cells that we transduced with TCRs specific for human Mesothelin³. We also included more detailed information regarding T cell/tumor cell ratios to the figure legend.

3. mRNA transfer using nanocarriers (in vivo experiments).

The authors apply nanocarrier technology in immunodeficient mouse models (NSG) and compare the efficiency of reprogrammed T cells with T cells redirected using retroviral gene transfer.

1. CD19+ Leukemia mouse model. The authors present that CD19-specific CAR T cells, regardless of the type of gene transfer technology, increase the survival of the tumor bearing mice. Using three examples representing a larger group of animals (Fig 4D), the authors show the CAR expression on CD8+ T cells at different time points of the animal experiment before and after infusion of nanocarriers loaded with CD19 CAR mRNA. In principle, the specificity of the nanocarrier technology should also be asked here. Both CD4 and CD8 T cells should be examined for CAR expression. It would definitely not be an advantage if the CARs or TCRs were expressed in cell types other than the desired ones.

A simple mouse model would be best, where the authors insert GFP mRNA into CD8-specific nanocarriers and inject the carriers into immunocompetent mice. Subsequently, immune cells (T-, B-, NK, DC, . . .) and certain tissue cells (lung, liver, brain, kidney, . . .) must be tested for GFP expression. Only when the specificity of the approach has been clarified does it make sense to think about applying this technology elsewhere.

To examine how exclusively targeting can confine nanoparticle interactions to circulating T cells, we employed the fully immunocompetent Ai14 reporter mouse⁴. In this genetically modified model, all cells contain a loxP-flanked STOP cassette preventing transcription of a CAG promoter-driven tdTomato protein. Only cells that are successfully transfected with mRNA encoding Cre recombinase (Cre) would excise the loxP-flanked STOP cassette, resulting in permanent tdTomato transcription and subsequent strong, amplified tdTomato expression. We first measured whole-organ fluorescence in Ai14 mice following a single injection of CD3-targeted (or isotype control-functionalized) nanoparticles carrying Cre mRNA. The highest gene expression mediated by non-targeted particles were found in the liver, whilst lymphocyte-targeted nanocarriers induced gene transfer mainly in the spleen, lymph nodes, and thymus (Fig. C below, new **Fig. 5a, b**). Much lower levels of dtTomato signals were detected in other CD45+ (immune) subtypes, such as macrophages (3.2% ± 1.5%), B cells (1.1% ± 0.9%), neutrophils (0.3% ± 0.2%), and dendritic cells (1.9% ± 0.8%).

4. ROR1+ Prostate tumors

As in the previous mouse experiment: Please first prove the specificity of your gene transfer method (CD4+, CD8+,...).

The ability of nanoparticles to specifically target circulating T cells is independent of their mRNA cargo. We demonstrate in the new Fig. 5 of our revised manuscript effective T cell targeting in fully immunocompetent mice. This specificity of in situ transfection does not change when switching from a leukemia-specific CD19 CAR to a prostate tumor-specific CAR.

5. HBV-specific TCR T cells

Unfortunately, this approach also lacks proof of specificity for presented gene transfer method.

See answer to question 4.

6. The toxicity studies were shown in Figures 7 and 8. But, the initial aim is to investigate how T cells react to multiple administration of nanocarriers. Does repeated administration of the nanocarrier lead to apoptosis of the T cell? The authors should first answer the more obvious questions for specificity of presented technology and for T cell biology.

To measure whether repeated nanoparticle infusion compromises the viability of in situ transfected T cells, Ai14 reporter mice (see Fig. C in Question 3) were injected intravenously with three daily doses (day 1-3) of nanoparticles loaded with 15 μg mRNA encoding nuclear localization signal (NLS)-Cre. Nanoparticles were targeted to mouse T cells using a full-length anti-CD3 Mu1gG2a designed as LALAPG variants to ablate Fc receptor binding and complement activation. Forty-eight hours after the final injection (day 5), single cell suspensions of spleens were labelled with antibodies against CD45 and T cell markers (CD4+ or CD8+), washed twice with cold cell staining buffer (Biolegend catalog # 420201) and then resuspended in Annexin V binding buffer at a concentration of 1×10^6 cells/ml. The apoptosis/necrosis markers used were Annexin A5 (Biolegend catalog # 640953, 1:20) and 7-AAD (Biolegend catalog # 420404, 1:100). Data were collected using a BD FACSymphony analyzer running FACSDIVA software (Beckton Dickinson) and summarized as new **Supplementary Fig. 2** (Fig. D).

Based on this study, we could not measure any effects on viability following repeated infusions of T-cell targeted mRNA nanocarriers.

Fig. D: In situ transfection of T cells does not compromise their viability. (a) FACS profiles of Annexin V/7-AAD staining of splenocytes using the following gates:

CD4+ T cells, nanoparticle-transfected: CD45+ dtTomato+ CD4+

CD4+ T cells, untransfected: CD45+ dtTomato- CD4+

CD8+ T cells, nanoparticle-transfected: CD45+ dtTomato+ CD8+

CD8+ T cells, untransfected: CD45+ dtTomato- CD8+

(b) Bar graphs displaying the mean % \pm SD of live, early apoptotic, late apoptotic, and necrotic cells. Pairwise differences in fluorescent intensities between the groups were statistically analyzed with the Wilcoxon rank-sum test (two-sided).

Summary: We appreciate the constructive criticism provided by this reviewer, and hope that our responses appropriately address the issues raised. We believe that the changes we made in response to them have substantially improved our manuscript.

References

1. Moffett, H.F., *et al.* Hit-and-run programming of therapeutic cytoreagents using mRNA nanocarriers. *Nat Commun* **8**(2017).
2. Gade, T.P., *et al.* Targeted elimination of prostate cancer by genetically directed human T lymphocytes. *Cancer Res* **65**, 9080-9088 (2005).
3. Stromnes, I.M., *et al.* T Cells Engineered against a Native Antigen Can Surmount Immunologic and Physical Barriers to Treat Pancreatic Ductal Adenocarcinoma. *Cancer Cell* **28**, 638-652 (2015).
4. Kauffman, K.J., *et al.* Rapid, Single-Cell Analysis and Discovery of Vected mRNA Transfection In Vivo with a loxP-Flanked tdTomato Reporter Mouse. *Mol Ther Nucleic Acids* **10**, 55-63 (2018).

REVIEWERS' COMMENTS

Reviewer #1 (Remarks to the Author):

The authors have nicely revised the paper according to our suggestions. The paper should be published ASAP.

- Dan Peer

Reviewer #2 (Remarks to the Author):

It is great to see the addition of new data, especially the work in immunocompetent mice. I would consider the authors have tried their best to answer all my queries and have addressed most of my questions well. I consider this work novel and can influence the field significantly.

I have a few minor suggestions for the authors to consider:

1. The authors have addressed my questions on how the nanocarriers were broken down and how the mRNA incorporated into the T cells. I would consider it is beneficial if the authors mention these mechanisms in the manuscript, as the audience might have the same questions as what I had.
2. I still have concerns on the toxicity. However, I do acknowledge that the authors have tried their best to address my concerns. The use of a rat model without the presence of the targeted antigen is not sufficient to address the toxicities. But as the authors pointed out that rodents cannot faithfully recapitulate T cell-mediated toxicity. I think it will be beneficial to carry out more discussions around the toxicity concerns in the discussion section rather than considering the method safe.
3. There are some general confusions on the methods for NP infusion and human T cells. For example, in Figure 3, the authors used activated T cells. But in the rest of the manuscript, the authors did not mention whether the T cells adoptively transferred into mice were pre-activated. I assume they were not activated. Please specify in either figure legend or method section. In addition, in some figures, the authors used anti-CD8 such as Figure 3, but in others, used CD3, such as Figure 5. Please be clear which NP you use for each figure. I would recommend the authors to add a section in the "methods" section on how you did the treatments. i.e. injected human T cells 1st, then nanoparticle a few hours later? Or in bilateral veins? This is one of the most important information for this manuscript.
4. In some figures, the authors used anti-CD8 such as Figure 3, but in others, used CD3, such as Figure 5. Please be clear which NP you use for each figure. Is there a particular reason that you swapped anti-CD8 and CD3? Do CD4 cells play a role in this system (just for my interest, no additional experiment required)?
5. From page 10 onwards, nearly all the figures used the wrong number, i.e. On page 10, Figure 5 should be Figure 7, page 13 figure 6a should be figure 8a. Please correct them all.
6. Figure 10a, all the kidney sections look like pancreatic islets to me.
7. Figure 4a, "D0", "O" is missing under "human T cells" and "NP" infusion.
8. Please update the reference on page 6 "two already FDA- approved cancer therapies". There was another CAR T cell therapy being FDA approved in 2020.

Reviewer #3 (Remarks to the Author):

"Targeted IVT mRNA nanocarriers program circulating T cells with disease-specific CARs or TCRs"

The authors have provided a convincing answer to the questions asked, thus ensuring greater clarity of the method presented. The manuscript is now more convincing in its meaning and meets

the expected requirements. Based on new data from in vitro and in vivo experiments, the authors are able to characterize the method more precisely and to determine the specificity of the nanoparticle-mediated CAR-T cell generation. We are now also learning the influence of repeated administration of nanoparticles on the viability of T cells.

The manuscript is well written and is important for the further development of CAR-T cell immunotherapy, as virus-mediated gene transfer is an extremely complex procedure and requires a large amount of logistics. The method presented would simplify the procedure if it complies with the clinical safety requirements.

Reviewer #1 (Remarks to the Author):

1. The authors have nicely revised the paper according to our suggestions. The paper should be published ASAP.

Reviewer #2 (Remarks to the Author):

1. The authors have addressed my questions on how the nanocarriers were broken down and how the mRNA incorporated into the T cells. I would consider it is beneficial if the authors mention these mechanisms in the manuscript, as the audience might have the same questions as what I had.

We added this information to our revised manuscript.

2. I still have concerns on the toxicity. However, I do acknowledge that the authors have tried their best to address my concerns. The use of a rat model without the presence of the targeted antigen is not sufficient to address the toxicities. But as the authors pointed out that rodents cannot faithfully recapitulate T cell-mediated toxicity. I think it will be beneficial to carry out more discussions around the toxicity concerns in the discussion section rather than considering the method safe.

We revised our discussion and are now adding the following paragraph:

“Prior to clinical studies in humans, we will extend our collaboration with the NCL to confirm safety of nanoparticles in a large-animal species, with reference to FDA regulations for nanomedicine. In contrast to therapeutics that are already established in the clinic, such as small molecules or antibodies, CAR-programming nanoparticles are multi-component three-dimensional constructs that require a reproducible manufacturing process to reliably achieve the intended physicochemical characteristics, biological behaviors, and pharmacological profiles. The safety and efficacy of such nanomedicines can be influenced by minor variations in many parameters and need to be carefully monitored, particularly in the context of targeting to unintended sites and potential toxicities. Furthermore, nanomedicines require additional developmental and regulatory considerations compared with conventional medicines. Only a few facilities with the requisite degree of expertise are currently operational in the United States.”

3. There are some general confusions on the methods for NP infusion and human T cells. For example, in Figure 3, the authors used activated T cells. But in the rest of the manuscript, the authors did not mention whether the T cells adoptively transferred into mice were pre-activated. I assume they were not activated. Please specify in either figure legend or method section. In addition, in some figures, the authors used anti-CD8 such as Figure 3, but in others, used CD3, such as Figure 5. Please be clear which NP you use for each figure. I would recommend the authors to add a section in the “methods” section on how you did the treatments. i.e. injected

human T cells 1st, then nanoparticle a few hours later? Or in bilateral veins? This is one of the most important information for this manuscript.

For all *in vitro* experiments in Figure 3, T cells needed to be activated with anti-CD3/CD28 coated beads as part of the transduction protocol with CAR-encoding retrovirus (which needs proliferating T cells). For all *in vivo* experiments, T cells were not pre-stimulated, and we are now specifically adding this information to the main text. Regarding the anti-mouse CD3 antibody, we revised our methods section, accordingly:

“Anti-CD3 murine IgG2a LALA-PG and nonbinder control (used only for *in vivo* experiments in immunocompetent mice/ Figure 6)”.

Regarding the timing of T cell- versus nanoparticle-infusion, this information is shown as a timeline (panel a of Figs. 4 and 7) for all *in vivo* experiments. We reconstituted mice with human T cells and (D-2), and started infusing nanoparticles 2 days later (D 0).

4. In some figures, the authors used anti-CD8 such as Figure 3, but in others, used CD3, such as Figure 5. Please be clear which NP you use for each figure. Is there a particular reason that you swapped anti-CD8 and CD3? Do CD4 cells play a role in this system (just for my interest, no additional experiment required)?

We only used CD3 for *in vivo* experiments in immunocompetent C57BL/6 mice. In the presence of a functional immune system it is crucial to functionalize nanoparticles with a targeting ligand that lacks a functional Fc. Since the deglycosylation kit we used to deactivate anti-human CD8 antibody (full length, purchased from BioXcell) did not work well on mouse antibodies, we decided to design our own construct, based on the 2C11 scFv (anti-mouse CD3 epsilon chain) with a “LALA” mutation, that abolishes Fc-mediated effector function. This construct was then custom-expressed for us by ATUM biosciences.

5. From page 10 onwards, nearly all the figures used the wrong number, i.e. On page 10, Figure 5 should be Figure 7, page 13 figure 6a should be figure 8a. Please correct them all.

We corrected figure numbers in the revised manuscript.

6. Figure 10a, all the kidney sections look like pancreatic islets to me.

Thank you for pointing this out. We corrected this mistake and are showing the kidney histology in the revised Figure 10.

7. Figure 4a, “D0”, “0” is missing under “human T cells” and “NP” infusion.

We corrected the labelling of this figure.

8. Please update the reference on page 6 "two already FDA- approved cancer therapies". There was another CAR T cell therapy being FBS approved in 2020.

We added Tecartus (brexucabtagene autoleucel) as third FDA-approved CAR T-cell therapy to our references

Reviewer #3 (Remarks to the Author):

1. The authors have provided a convincing answer to the questions asked, thus ensuring greater clarity of the method presented. The manuscript is now more convincing in its meaning and meets the expected requirements. Based on new data from in vitro and in vivo experiments, the authors are able to characterize the method more precisely and to determine the specificity of the nanoparticle-mediated CAR-T cell generation. We are now also learning the influence of repeated administration of nanoparticles on the viability of T cells. The manuscript is well written and is important for the further development of CAR-T cell immunotherapy, as virus-mediated gene transfer is an extremely complex procedure and requires a large amount of logistics. The method presented would simplify the procedure if it complies with the clinical safety requirements.